# Structural basis of human full-length kindlin-3 homotrimer in an auto-inhibited state

**Wenting Bu**[1]**, Zarina Levitskaya**[1]**, Zhi Yang Loh**[1]**, Shengyang Jin**[1,2]**, Shibom Basu**[3]**, Rya Ero**[1]**, Xinfu Yan**[1]**, Meitian Wang**[3]**, So Fong Cam Ngan**[1]**, Siu Kwan Sze**[1]**, Suet-Mien Tan**[1]***, Yong-Gui Gao**[1,2,4]*

**1** School of Biological Sciences, Nanyang Technological University, Singapore, Singapore, **2** NTU Institute of Structural Biology, Nanyang Technological University, Singapore, Singapore, **3** Swiss Light Source, Paul Scherrer Institute, Villigen, Switzerland, **4** Institute of Molecular and Cell Biology, Agency for Science, Technology and Research, Singapore, Singapore

* ygao@ntu.edu.sg (YGG); smtan@ntu.edu.sg (SMT)

## Abstract

Kindlin-1, -2, and -3 directly bind integrin β cytoplasmic tails to regulate integrin activation and signaling. Despite their functional significance and links to several diseases, structural information on full-length kindlin proteins remains unknown. Here, we report the crystal structure of human full-length kindlin-3, which reveals a novel homotrimer state. Unlike kindlin-3 monomer, which is the major population in insect and mammalian cell expression systems, kindlin-3 trimer does not bind integrin β cytoplasmic tail as the integrin-binding pocket in the F3 subdomain of 1 protomer is occluded by the pleckstrin homology (PH) domain of another protomer, suggesting that kindlin-3 is auto-inhibited upon trimer formation. This is also supported by functional assays in which kindlin-3 knockout K562 erythroleukemia cells reconstituted with the mutant kindlin-3 containing trimer-disrupting mutations exhibited an increase in integrin-mediated adhesion and spreading on fibronectin compared with those reconstituted with wild-type kindlin-3. Taken together, our findings reveal a novel mechanism of kindlin auto-inhibition that involves its homotrimer formation.

## Introduction

Integrin-mediated cell–cell and cell–extracellular matrix interactions are crucial for many physiological and pathological processes. Recently, the 4.1 protein, ezrin, radixin and moesin (FERM)-domain–containing kindlin family of proteins has emerged as key players involved in integrin activation. The 3 members are kindlin-1, kindlin-2, and kindlin-3 [1] (Fig 1A). The importance of kindlins is underscored by their link to Kindler syndrome (a disorder within the spectrum of hereditary epidermolytic blistering diseases), leukocyte adhesion deficiency (LAD) III (a hereditary primary immunodeficiency with platelet dysfunction disease), and cancer [2,3]. The sequence homology between kindlin-1 and kindlin-2 is 62%, and those between kindlin-1 and kindlin-3, as well as kindlin-2 and kindlin-3 are 54% and 52%, respectively (44% for all 3). In addition to sequence similarity, all kindlins share atypical FERM domain organization comprising the F1, F2, and F3 subdomains (Fig 1A and S1 Fig). Studies

pdb7C3M/pdb). Cross-linking mass spectrometry data are available via ProteomeXchange with identifier PXD019110 (ftp://ftp.pride.ebi.ac.uk/pride/data/archive/2020/06/PXD019110).

**Funding:** This work was supported by the Tier II grants MOE2017-T2-1-106 (YGG) and MOE2016-T2-1-021 (SMT) from the Ministry of Education (MOE) of Singapore. This research was also supported by the National Research Foundation Singapore under its Open Fund - Individual Research Grant (MOH-000218) (SMT) and administered by the Singapore Ministry of Health's National Medical Research Council. The funders had no role in study design, data collection and analysis, decision to publish, or preparation of the manuscript.

**Competing interests:** The authors have declared that no competing interests exist.

**Abbreviations:** AFA, triple mutations Q471A, A475F, and S478A; BiFc, bimolecular fluorescence complementation; CD, circular dichroism; DSSO, disuccinimidyl sulfoxide; *E. coli*, *Escherichia coli*; EI, expression index; EM, electron microscopy; e-YFP, enhanced yellow fluorescent protein; F3Δ, F3 domain deleted; F-actin, filamentous actin; Fc, calculated structure factor; FERM, 4.1 protein, ezrin, radixin and moesin; Fo, observed structure factor; GAPDH, glyceraldehyde 3-phosphate dehydrogenase; GFP, green fluorescent protein; GM, geo-mean fluorescence; GP, percent gated positive; HA, human influenza hemagglutinin; HEK, human embryonic kidney; I.P., immunoprecipitation; IgG, immunoglobulin; ILK, integrin-linked kinase; ITC, isothermal titration calorimetry; Kd, dissociation constant; KO, knockout; LAD, leukocyte adhesion deficiency; mAb, monoclonal antibody; MS, mass spectrometry; NLS, nuclear localization sequence; PH, pleckstrin homology; PTB, phosphotyrosine-binding; PtdIns(3,4,5)P3, phosphatidylinositol (3,4,5)-trisphosphate; SAD, Se-single-wavelength anomalous diffraction; Se, selenine; SEC-MALS, size-exclusion chromatography multiangle light scattering; Sf9, *Spodoptera frugiperda* 9; WT, wild type; YFP, yellow fluorescent protein.

have revealed that the poly-lysine-containing loop in the F1 subdomain of kindlins is important for interactions with membrane lipids [4,5]. The F2 subdomain is bisected by a pleckstrin homology (PH) domain, which is involved in binding membrane phosphatidylinositol lipids [6]. In particular, the PH domain has an essential role for kindlin-3 in integrin-mediated B cell adhesion and migration [6]. The F3 subdomain binds to the highly conserved membrane distal NxxY/F motif in the integrin β cytoplasmic tail [7].

Integrins are bidirectional signal transducers that mediate both inside-out and outside-in signaling [8]. Integrin activation involves conformational changes [9], which are regulated by interaction between their cytoplasmic tails and cytoplasmic proteins [10]. Kindlins cooperate with talin to positively regulate integrin activation [11]. Kindlins do not directly interact with talin. Instead, both molecules can simultaneously associate with a single integrin β cytoplasmic tail [12]. There are 2 well-defined motifs that are part of a canonical recognition sequence for phosphotyrosine-binding (PTB) domains in the β tails of integrin: the membrane proximal and distal motifs NPxY and NxxY (x: other amino acids), respectively [13]. Talin binds the membrane proximal NPxY motif, whereas kindlin binds the membrane distal NxxY motif [11]. Despite clear functional evidence of kindlins serving as positive regulators of integrin activation, the detailed mechanism remains to be determined.

Structural characterization of kindlins would provide important insights into the mechanisms by which kindlins interact with integrins and other binding partners. Despite a number of biochemical and structural studies on kindlins, their full-length structure is still unknown [14–18]. Nonetheless, the structure of a truncated mouse kindlin-2 suggests a mechanism by which kindlin-2 forms homodimer that promotes integrin clustering [18]. However, it should be noted the strategy employed in the aforementioned study in order to facilitate crystallization involved removing the PH domain, resulting in artificially linking the 2 sides of the F2 domain as well as deleting the long loop within the F1 domain. These modifications could potentially generate global perturbations in the native kindlin-2 structure, which, in turn, may explain the extremely slow process of dimerization reported in the same study.

Here, we report the crystal structure of human full-length kindlin-3. Structural, biophysical, and functional studies provide evidence of kindlin-3 homotrimer population both in vitro and in vivo. Unlike kindlin-3 monomer, the trimer does not bind integrin β1A cytoplasmic tail, suggesting an auto-inhibitory mechanism mediated by trimer formation. This is supported by cell-based functional studies in which cells expressing a mutant kindlin-3 containing disrupting mutations in its trimer interface exhibited overt β1 integrin-mediated cell adhesion and spreading as compared with cells expressing wild-type kindlin-3. Taken together, our data reveal a regulatory mechanism of kindlin-3 involving its trimer formation in integrin signaling.

## Results

### Structure of kindlin-3 monomer

To gain mechanistic insights into the functions of kindlins, characterization of their full-length structure at atomic resolution is required. Full-length kindlins contain disordered regions that are intrinsically recalcitrant to crystallization. Through extensive and time-consuming efforts of large-scale preparation of highly homogeneous kindlins and subsequent crystallization trials, we were ultimately able to crystallize human full-length kindlin-3 that was expressed as a single monomeric species in bacterial *Escherichia coli* BL21 system. Consequently, we determined the first structure of human full-length kindlin-3, using the selenine (Se)–single-wavelength anomalous diffraction (SAD) phasing method in combination with Phenix program (Paul Adams, Randy Read, Jane & Dave Richardson, Tom Terwilliger; https://www.phenix-online.org/)

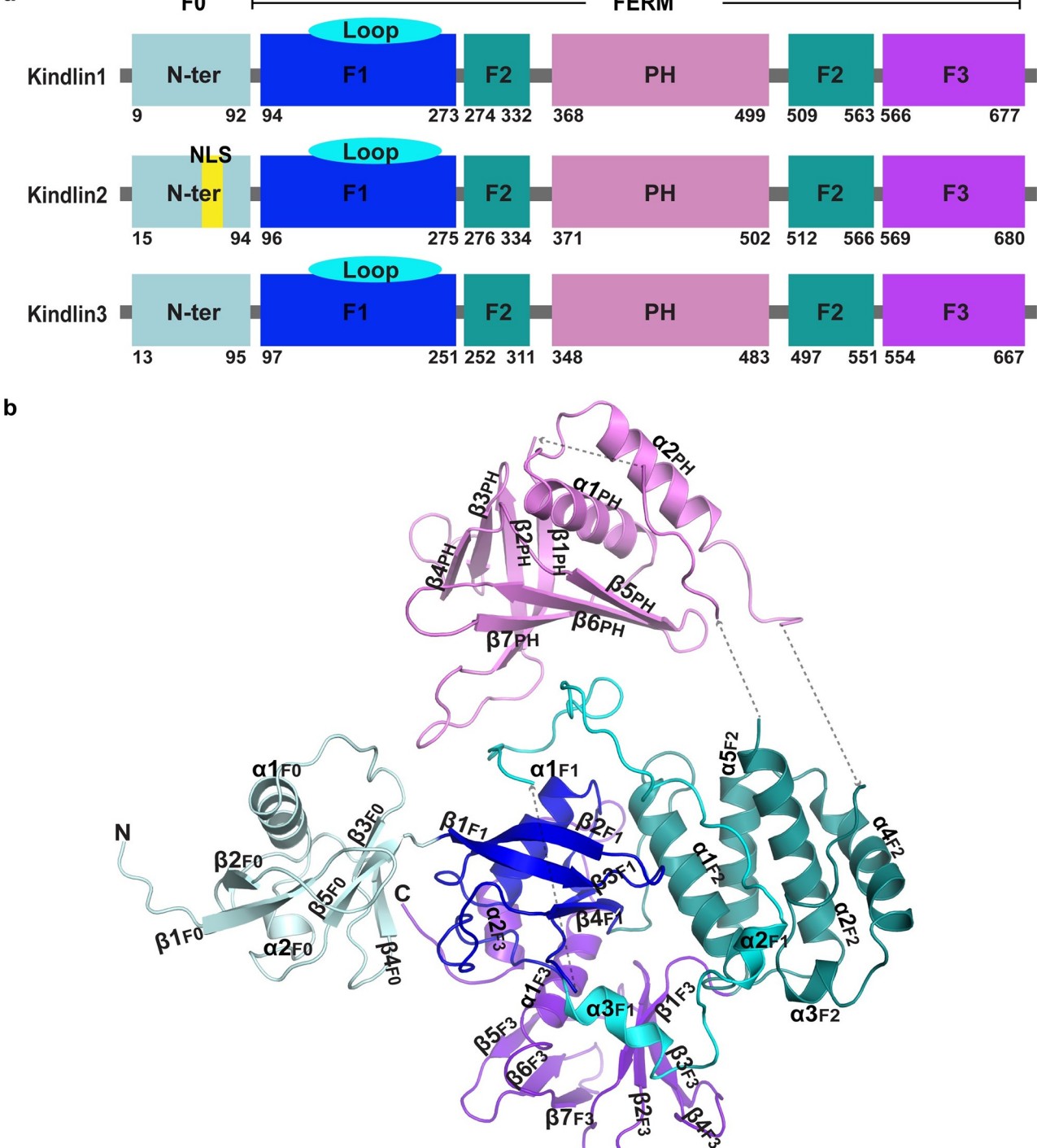

**Fig 1. Domain organization of kindlins and the overall structure of human full-length kindlin-3.** (**a**) Domain organization of human kindlins. Domains F0, F1, the F1 inserted loop, domains F2, PH, and F3 are colored pale cyan, blue, aquamarine, deep teal, violet, and purple blue, respectively. Domains and the F1 loop are colored the same throughout the paper. (**b**) Crystal structure of human full-length kindlin-3. The secondary structure elements are labeled. FERM, 4.1 protein, ezrin, radixin and moesin; NLS, nuclear localization sequence; PH, pleckstrin homology.

[19,20] (Fig 1B, S2–S4 Figs). Unexpectedly, we observed 3 molecules of kindlin-3 forming a homotrimer in an asymmetric unit of the crystal with each protomer dovetailing the next. Each kindlin-3 monomer adopts a cloverleaf-like overall conformation. Except for the F2 domain, which is an α-bundle, the other 4 domains are composed of an α/β barrel. The overall structure of human kindlin-3 monomer is similar to that of truncated mouse kindlin-2 [18]; however, the relative positioning of the PH domain can be resolved in the former.

The N-terminal F0 domain adopts a ubiquitin-like fold comprising 5 β-stands and 2 helices ($\alpha1_{F0}$ and $\alpha2_{F0}$), forming a classical α/β barrel. The F1 domain contains an α/β barrel similar to that of the F0 domain, but it has a long loop (residues 145–228) that is conserved amongst kindlins and functionally important [5]. Except for residues 146–174, most components of this 83-residues-long region can be modeled, with the loop 181–187, the loop 196–207, and its subsequently connecting helices $\alpha2_{F1}$ and $\alpha3_{F1}$ interacting with domains PH, F2, and F3, respectively (Fig 1B). Similar to kindlins, talin also contains a F1 loop but shorter in length (31 residues). The positively charged F1 loop binds the negatively charged membrane, which is required for the regulatory functions on integrins [5,21,22]. The F1 domain resides at the center of the kindlin-3 structure with the F1 loop wrapped around its α/β barrel and interacting with other domains. This arrangement may contribute, in part, to the functional difference between kindlins and talin, given the different lengths of their F1 loops [22].

The F2 domain is composed of 5 helices that form a compact α-bundle (Fig 1B). The first 2 helices $\alpha1_{F2}$ and $\alpha2_{F2}$ cross one another. The remaining 3 helices form an "arch" arrangement and pack against the $\alpha2_{F2}$ helix. The PH domain, which is inserted between $\alpha2_{F2}$ and $\alpha3_{F2}$, comprises 7 β-strands and 2 α-helices. The 7 β-strands are assembled in 2 perpendicular antiparallel β-sheets (the first 4 orthogonal to the remaining 3), which forms a compact β-barrel. One end of this barrel is blocked by the C-terminal α-helices ($\alpha1_{PH}$ and $\alpha2_{PH}$), and the opposite end of the barrel is rich in positively charged residues and exposed, suggesting that it may be involved in membrane lipid binding [17]. In parallel with these 2 α-helices ($\alpha1_{PH}$ and $\alpha2_{PH}$), a loop between $\alpha2_{F2}$ and $\beta1_{PH}$ was modeled at the N-terminus of the PH domain (Fig 1B). Given the limited electron density map and a lack of a molecular model, this loop appears to be a helix interacting with $\alpha2_{PH}$ to stabilize the entire domain (S5 Fig). The PH domain, widely distributed in other protein families (the 11th most common domain family in human proteome), has been implicated in diverse biological processes, including intracellular signaling, cytoskeleton constitution, intracellular membrane trafficking, and membrane phospholipids modification [23,24].

The F3 domain, located at the opposite side of the PH domain, is composed of 2 perpendicular antiparallel β-sheets ($\beta1_{F3}$–$\beta4_{F3}$ against $\beta5_{F3}$–$\beta7_{F3}$). These form a β-barrel followed by 2 α-helices that block one end of the barrel. These 2 α-helices are held in position by contacts with both the F3 barrel and the F1 domain (Fig 1B). The F3 domain also contains a PTB-like region involving a cleft surrounded by the last α-helix and the β-sheet ($\beta5_{F3}$-$\beta7_{F3}$), which is the site for integrin β cytoplasmic tail binding.

## Kindlin-3 homotrimer

A salient feature of the kindlin-3 crystal structure is the presence of 3 molecules of kindlin-3 forming a homotrimer in an asymmetric unit (Fig 2A–2C). Kindlin-3 trimer adopts a triangular prism-like structure along the noncrystallographic 3-fold symmetry axis with the following dimensions: an outer-side length of approximately 110 Å, an inner cavity width of approximately 60 Å, and a height of approximately 90 Å. Our crystal structure data of kindlin-3 trimer were obtained using homogenous kindlin-3 monomers expressed and purified from *E. coli*. To exclude the possibility of crystallization artefacts, it is important to determine

whether kindlin-3 trimer exists in eukaryotic cells. To this end, human full-length kindlin-3 was expressed and purified from a clonal isolate of *Spodoptera frugiperda* Sf21 cells (Sf9) insect cells. We were able to isolate both kindlin-3 monomer and trimer forms, based on their molecular weights calibrated by analytical gel filtration chromatography (Fig 2D). Furthermore, the particle size of a kindlin-3 trimer was approximately 10–12 nm based on negative staining electron microscopy (EM) (S6 Fig), which is comparable in size with that observed in the crystal structure. In addition, we performed disuccinimidyl sulfoxide (DSSO: a mass spectrometry-cleavable crosslinker) crosslinking of kindlin-3 monomer and trimer followed by analytical gel filtration and mass spectrometry (MS) analysis. Compared with DSSO-untreated monomer kindlin-3, DSSO-treated monomer kindlin-3 exhibited a clear band above 200k Da (approximately equivalent to molecular weight of 3 folds of monomeric kindlin-3) on denaturing SDS-PAGE and a clear curve shift forward in analytical gel filtration profile (S7A and S7B Fig). This indicated that monomer kindlin-3 was crosslinked to trimer kindlin-3. In addition, DSSO-treated trimer kindlin-3 also exhibited a clear band above 200 k Da on denaturing SDS-PAGE (S7C Fig). Both 2 crosslinked trimeric bands were sent for MS fragmentation analysis. There were 4 pairs of identified crosslinked residues, among which 3 have very high confidence (S8 Fig **and** S1 Data). Interestingly, 2 lysine residues from $\alpha 1_{PH}$ (LAS<u>K</u>GR) of one protomer and $\beta 2_{F3'}$ (<u>K</u>DEILGIANNR) of another protomer were crosslinked by DSSO, and it is consistent with their near distance in the crystal structure of the kindlin-3 trimer. It should be noted that circular dichroism (CD) spectra analysis of kindlin-3 monomers from both expression systems revealed minimal to no difference in terms of secondary structures (S9 Fig).

The crystal structure of kindlin-3 trimer also reveals that the additional $\alpha 2_{PH}$ helix in the PH domain of protomer A directly wedges into a cleft involving the 2 C-terminal helices ($\alpha 1_{F3}$ and $\alpha 2_{F3}$) and the β-sheet ($\beta 5_{F3}$-$\beta 7_{F3}$) in the F3 domain of protomer B, thereby forming extensive contacts. The residues Thr467, Ser468, Val470, Gln471, Ala475, and Ser478 in $\alpha 2_{PH}$ of protomer A interact with Ser595, Gln599, Trp600, Ser642, Thr643, Leu656, and Gln659 located in $\beta 5_{F3}'$ and its preceding loop, and $\alpha 1_{F3}'$ of F3 domain in protomer B (the prime $'$ refers to protomer B in the trimer), respectively (Fig 2C). In addition, the C-terminus of $\alpha 1_{PH}$ and its subsequent loop in the PH domain pack against the 2 C-terminal helices and their connecting loop in the F3 domain of protomer B, through H-bonding (Gln443 in protomer A with main chain of Phe640 in protomer B) and salt bridge (Arg447 in A-Asp653 in B) interactions (Fig 2C). Therefore, we proposed that the triple mutations Q471A, A475F, and S478A (henceforth referred to as AFA) could disrupt the protomer–protomer interface in kindlin-3 trimer. Indeed, the AFA mutant kindlin-3 forms only monomer in solution when expressed in insect cells (Fig 2D).

Following a similar manner of protomer–protomer assembly, the kindlin-3 trimer is formed by head-to-tail interactions between protomers in which the PH domain of protomer A interacts with the F3 domain of protomer B; likewise, the PH domain of protomer B interacts with the F3 domain of protomer C; and finally, the PH domain of protomer C interacts with the F3 domain of protomer A. Based on PISA program (E. Krissinel and K. Henrick; https://www.ebi.ac.uk/pdbe/pisa/picite.html) [25], a total surface of approximately 1,850 $\text{Å}^2$ is buried for each protomer in the trimer configuration.

## Kindlin-3 trimerization inhibits its integrin-binding ability

The structure of mouse kindlin-2 in complex with the integrin β1 cytoplasmic tail provides important insight into the mechanism by which kindlin interacts with integrin [18]. We therefore superimposed the structures of human full-length kindlin-3 and mouse kindlin-2-integrin

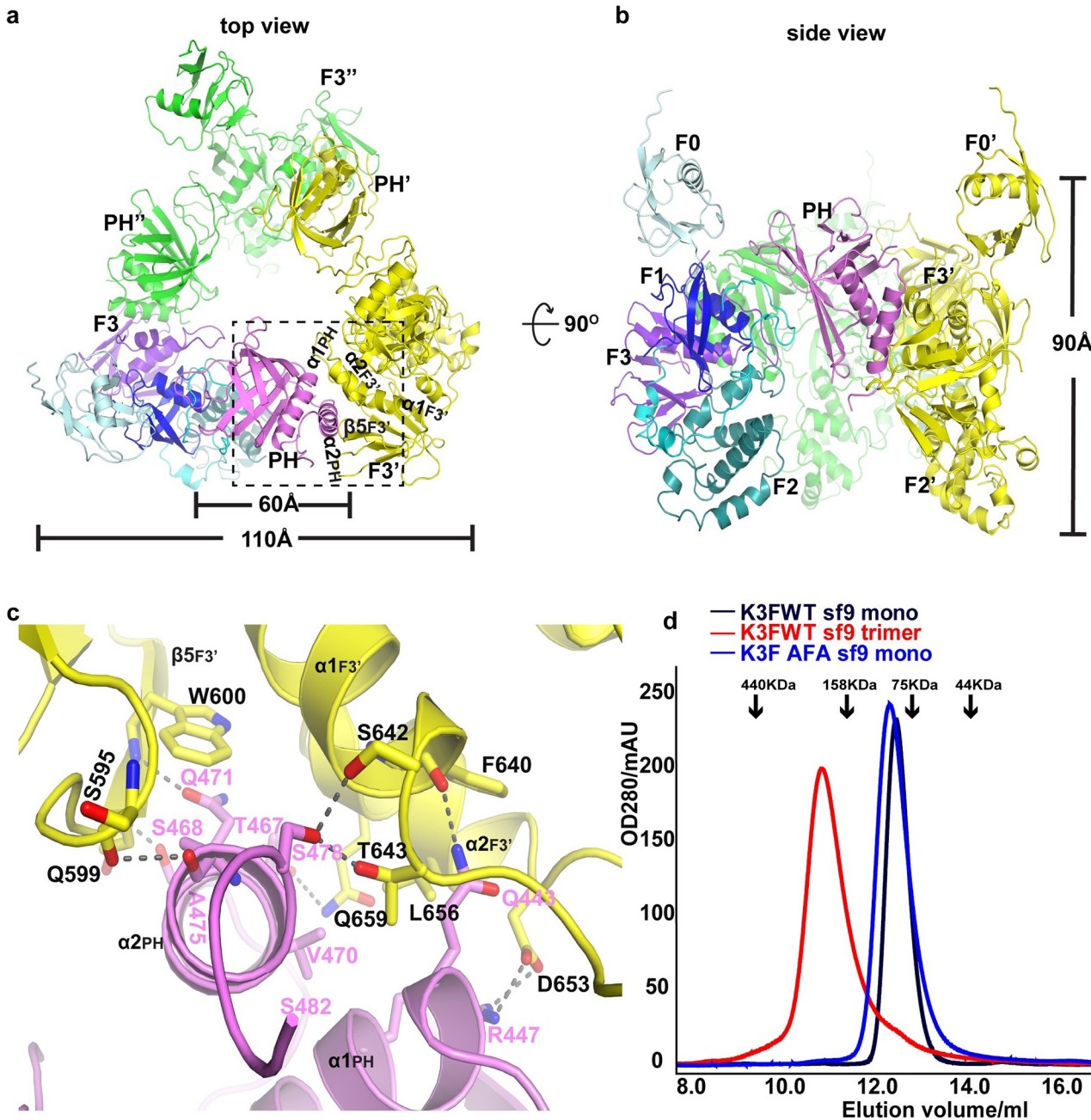

**Fig 2. Trimer formation of kindlin-3.** (**a**) "Top" view of the kindlin-3 homotrimer. One protomer is represented with domain shown in the same color scheme as in Fig 1A, the other 2 protomers are colored yellow and green, respectively. The prime ′ and double prime ″ refer to protomers B and C (same as below) in the trimer, respectively. (**b**) "Side" view of the kindlin-3 homotrimer. (**c**) Close-up view of the detailed interactions along the trimer interface between the PH domain of one protomer and the F3 domain of the neighboring protomer. The H-bonds and salt bridges are shown with dashed lines. (**d**) Analytical gel filtration chromatography profiles of native and mutant kindlin-3 purified from insect cells. K3FWT monomer (black) and trimer (red): the native kindlin-3 expressed in Sf9 insect cells could be prepared as both monomer and trimer; K3FAFA monomer (blue): the kindlin-3 with triple mutations Q471A, A475F, and S478A in trimer interface exhibits monomeric state. Note that molecular weight markers for analytical gel filtration chromatography are indicated by black arrows. AFA, triple mutations Q471A, A475F and S478A; PH, pleckstrin homology; Sf9, *Spodoptera frugiperda* 9; WT, wild type.

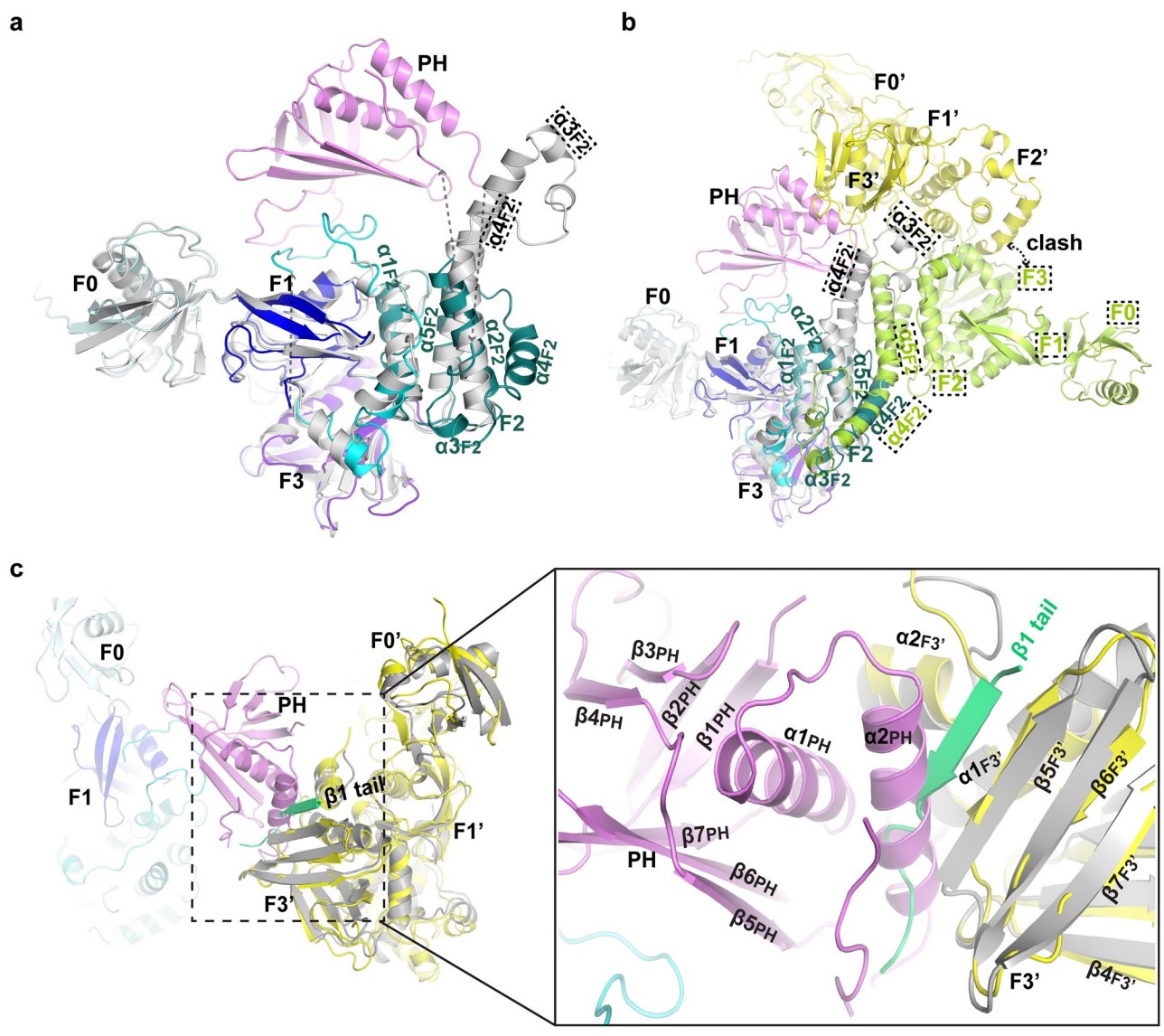

**Fig 3. Structural comparison of kindlin-2 and kindlin-3.** (**a**) Superposition of human full-length kindlin-3 and mouse PH-deleted kindlin-2 (gray) monomers. Note that the frames with dashed line indicate kindlin-2. (**b**) Superimposition of human kindlin-3 trimer and mouse PH-deleted kindlin-2 dimer. (**c**) The cleft in F3 subdomain that binds integrin β cytoplasmic tail is occluded by the helix α2$_{PH}$ (PH domain) of neighboring protomer in the kindlin-3 trimer. Integrin β tail bound to mouse PH-deleted kindlin-2 (gray) is colored lime green. PH, pleckstrin homology.

cytoplasmic tail complex. Although the F0, F1, and F3 domains occupy similar 3-D spaces in both structures (Fig 3A), there is a key difference in regard to the F3-integrin-binding pocket (Fig 3B). In the kindlin-3 trimer, the F3-integrin-binding site in each protomer is occluded by the α2$_{PH}$ helix of the PH domain from the neighboring protomer, suggesting that trimer formation could inhibit kindlin-3 binding to integrin β cytoplasmic tails (Fig 3C).

We therefore performed isothermal titration calorimetry (ITC) measurements to determine the binding affinities of integrin β1 cytoplasmic tail for kindlin-3 monomer and trimer that were purified from the insect cell expression systems, respectively. Although kindlin-3 monomer binds, albeit weakly with the dissociation constant (Kd) of approximately 200 μM, with integrin β1 cytoplasmic tail, kindlin-3 trimer does not show any detectable binding (S10A and S10B Fig).

The weak binding between kindlin-3 monomer and integrin β1 cytoplasmic tail is unlikely to be the result of nonoptimal ITC experimental conditions because ITC measurements using insect cell–purified human full-length kindlin-2 (monomer) and integrin β1 cytoplasmic tail yielded a Kd of approximately 13.4 μM (S10C Fig). Furthermore, the binding affinity between kindlin-3 monomer and integrin β1 cytoplasmic tail is consistent with that reported in a previous study [5], suggesting that the binding affinities of kindlins for the same integrin cytoplasmic tail are not similar. In addition, kindlin-2 trimer did not show any detectable binding with integrin β1 cytoplasmic tail (S10D Fig). Taken together, these data provide evidence that kindlin-3 trimer does not bind integrin β1 cytoplasmic tail.

## Trimer formation in kindlins

Although the insect cell expression system is a widely used eukaryotic system to overexpress mammalian proteins for structural and functional studies, we extended the study to determine whether kindlin-3 trimer can be detected in a mammalian cell expression system. To this end, we generated a stable human embryonic kidney (HEK) 293 cell line that expressed N-terminal His$_6$-tagged human full-length kindlin-3 containing a 3xGly linker, and purified kindlin-3 from these cells. In line with our previous data, we detected both trimer and monomer populations of kindlin-3 based on analytical gel filtration chromatography (Fig 4A). These data further validate the existence of kindlin-3 trimer in eukaryotic cells.

It is also of interest to determine if kindlin-1 and kindlin-2 can form homotrimers as well. We expressed and purified human full-length kindlin-2 using the insect cell expression system. Similar to kindlin-3, we detected both monomer and trimer populations of kindlin-2 based on analytical gel filtration chromatography (Fig 4B). Furthermore, the 2 populations (monomer and trimer) of kindlin-2 together with that of kindlin-3 were verified by size-exclusion chromatography multiangle light scattering (SEC-MALS) analysis (Fig 4C). These data suggest that trimer formation may be a conserved mechanism in kindlins.

In our analyses, we did not detect kindlin-3 or kindlin-2 homodimers. Given that kindlin-2 has been proposed to form a dimer that promotes integrin clustering [18], we compared the interface between protomers in mouse kindlin-2 and human kindlin-3 crystal structures. The dimer interface of mouse kindlin-2, in which the PH domain was removed and the 2 halves of F2 domain were artificially linked, involves an equivalent helical bundle from F2 of protomer A in the kindlin-3 structure (Fig 3A). This suggests that in full-length kindlins, the assembly of a dimer via the F2 interface may be severely limited by steric clashes (Fig 3B). In accordance, dimer formation of PH-deleted mouse kindlin-2 in solution was reported to occur only after several days [18]. Using similar conditions, dimer formation of human full-length kindlin-2 was also not detected by us, implying that the presence of the PH domain possibly interferes with dimer formation. However, given that the PH domain is linked to the F2 domain via flexible loops, the possibility that the PH domain orients away allowing dimer formation via the F2 interface cannot be completely ruled out, and perhaps more studies are needed.

It should also be noted that the human kindlin-3 structure reveals a similar F2 fold as the one observed in talin and the monomeric mouse kindlin-2 resulting from further mutations that prevent dimer formation (S11 Fig) but different than that of dimeric kindlin-2 (Fig 3B) [18,22], implying a conserved fold of the F2 domain in kindlins and talin, as well as an important role of the inserted PH domain in F2 in the oligomerization of kindlins.

## Kindlin-3 trimerization in cells

To exclude the possible artefacts arising from protein aggregation under high concentration conditions in vitro, we further designed a bead-based bimolecular fluorescence complementation

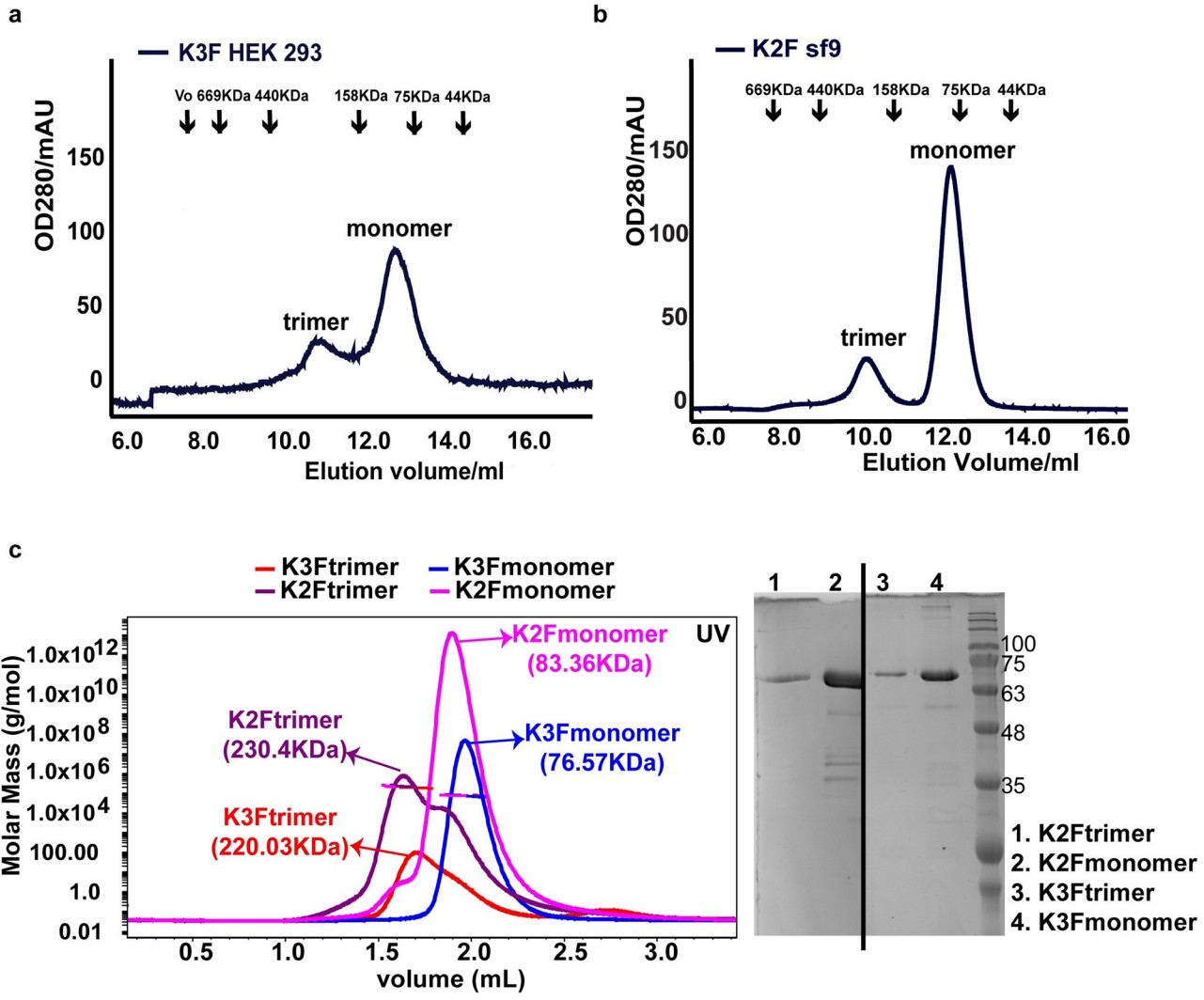

**Fig 4. Trimer formation in kindlins.** (a) Analytical gel filtration chromatography of kindlin-3 expressed and purified from HEK 293 mammalian cells indicates both monomeric and trimeric states in solution. (b) Analytical gel filtration chromatography of kindlin-2 expressed and purified from Sf9 insect cells indicates both monomeric and trimeric states in solution. (c) SEC-MALS analysis of kindlin-2 and kindlin-3 expressed and purified from Sf9 insect cells. Consistent with the analytical gel filtration chromatography, kindlin-3 forms both monomer (blue) and trimer (red) in solution. Likewise, kindlin-2 also forms monomer (pink) and trimer (purple). All samples were analyzed by SDS-PAGE before loading into the column. HEK, human embryonic kidney; SEC-MALS, size-exclusion chromatography multiangle light scattering; Sf9, *Spodoptera frugiperda* 9.

(BiFc) approach to detect kindlin-3 trimerization in small scale in transfected cells. Briefly, HEK 293T cells were transfected with 3 expression plasmids. One plasmid contains HA-tagged kindlin-3 cDNA, and each of the other 2 contains cDNAs of either the N-terminal half (M1-A155) or C-terminal half (D156-K239) of enhanced yellow fluorescent protein (eYFP) fused to the N-terminus of kindlin-3. In order to allow the 2 halves of the eYFP to come together forming an intact eYFP in the kindlin-3 trimer, we included a 12 Gly linker between each half of the eYFP and kindlin-3 to span the approximately 110-Å distance separating the F0 domains in the trimer configuration (Fig 5A). In cells transfected with all 3 plasmids, there will be kindlin-3 trimers with different combinations of each fusion protein, but only the combination consisting of one molecule each of HA-kindlin-3, N-eYFP-kindlin-3, and C-eYFP-kindlin-3 will fluoresce under excitation after immunoprecipitation. We also generated relevant expression plasmids for

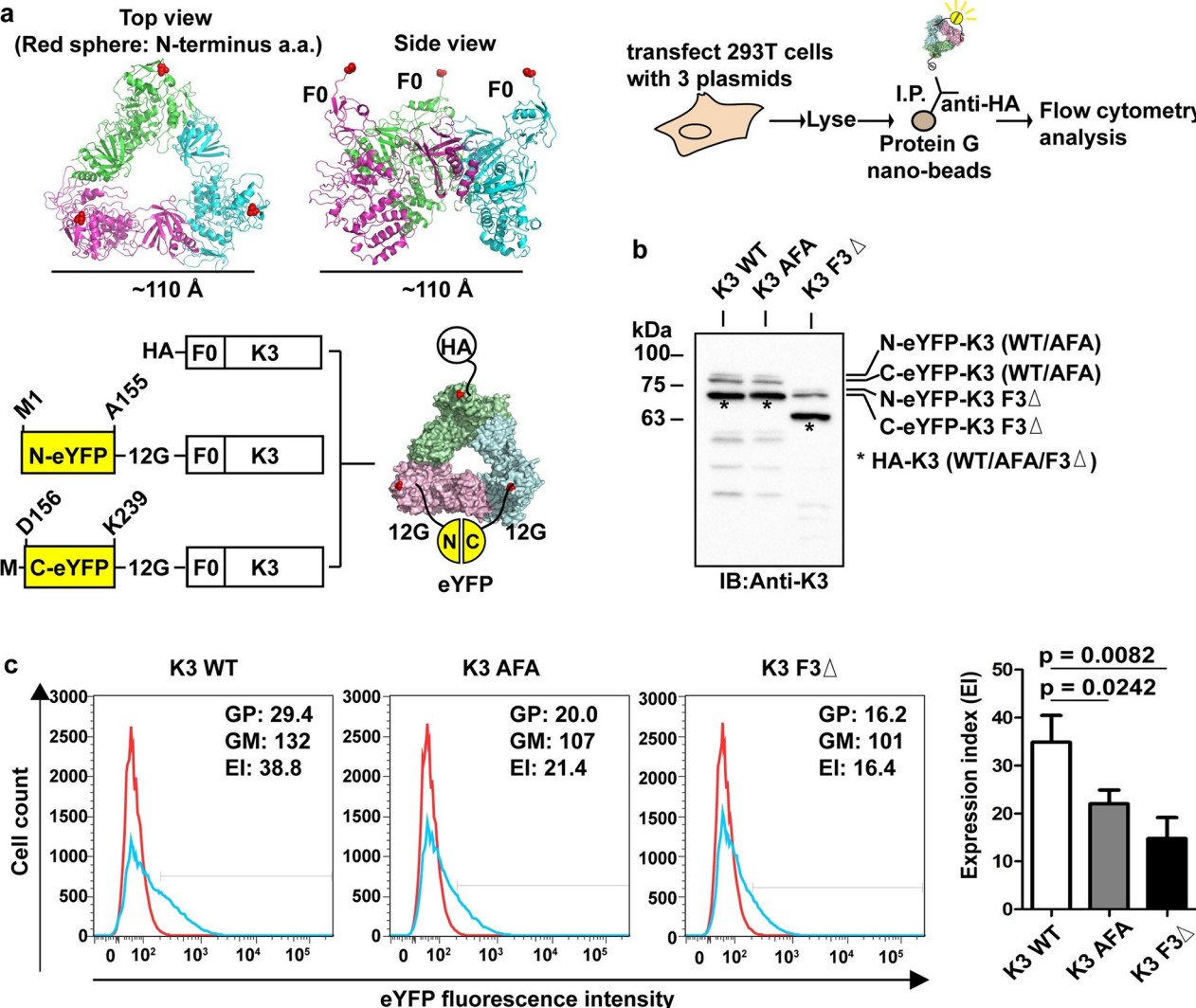

**Fig 5. Kindlin-3 trimerization in cells.** (**a**) An illustration of the design and analysis of kindlin-3 trimerization in cells using bead-based BiFc flow cytometry. The system made use of 3 expression plasmids, namely, HA-kindlin-3, N-eYFP(12G)-kindlin-3, and C-eYFP(12G)-kindlin-3. When an HEK 293T cell is transfected with all 3 expression plasmids, there will be trimers that have different combinations of monomers with different tags (i.e., HA, N-eYFP, C-eYFP). One of these combinations will contain one molecule each of HA-kindlin-3, N-eYFP(12G)-kindlin-3, and C-eYFP (12G)-kindlin-3. This combination can also facilitate BiFc via coming together of N-eYFP and C-eYFP. In order to detect this trimer population, immunoprecipitation can be performed using anti-HA antibody and protein G nanobeads, and YFP fluorescence can be detected by flow cytometry analysis. (**b**) Western blotting of 293T cells cotransfected with all 3 plasmids. Three protein bands corresponding to N-eYFP(12G)-kindlin-3, C-eYFP (12G)-kindlin-3, and HA-kindlin-3 were detected using anti-kindlin-3 antibody. (**c**) Left panel. Immunoprecipitation of kindlin-3 trimer using anti-HA antibody and protein G nanobeads followed by flow cytometry analysis to detect YFP signal as a result of BiFc. Red histogram: control IgG. Blue histogram: anti-HA antibody. EI = % GP × GM. Right panel. Expression index plot of 3 independent experiments. Two-tailed unpaired $t$ test was performed. Additional information on numerical values and data was provided in S2 Data. AFA, triple mutations Q471A, A475F, and S478A; BiFc, bimolecular fluorescence complementation; EI, expression index; eYFP, enhanced yellow fluorescent protein; F3Δ, F3 domain deleted; GFP, green fluorescent protein; GM, geo-mean fluorescence; GP, percent gated positive; HA, human influenza hemagglutinin; HEK, human embryonic kidney; IgG, immunoglobulin G; I.P., immunoprecipitation; YFP, yellow fluorescent protein.

kindlin-3 wild type (WT) and 2 variants to demonstrate trimer formation via the binding interface determined from our crystal structure (Fig 5B). The AFA mutant was designed to potentially disrupt the protomer–protomer interface. The F3 domain–deleted (F3Δ) mutant was chosen because F3 is required for trimer formation through its interaction with the PH domain.

Interestingly, the highest fluorescence signal was detected with WT kindlin-3, followed by the AFA and F3Δ mutants captured nanobeads, and the difference was statistically significant (Fig 5C). We inferred that AFA mutations diminished kindlin-3 trimer formation, but the disruption was lesser than that caused by F3 domain deletion. We noted that there is still fluorescence signal detected in the F3Δ sample. This could be due to nonspecific background binding to the beads and low level of self-association between N-eYFP and C-eYFP (a common observation in BiFc experiments) independent of kindlin-3 trimer formation.

## Disruption of kindlin-3 trimerization exhibits overt integrin activation and increased cell spreading

Our results suggest that, contrary to that of kindlin-2 [18], kindlin-3 oligomerization inhibits its function in terms of binding integrin β cytoplasmic tails. We reasoned that if kindlin-3 trimer represents an auto-inhibited state that is in dynamic equilibrium, as a mode of regulation, with kindlin-3 monomer in cells, then preventing trimer formation could induce overt integrin activation. To test this hypothesis, we performed functional studies in mammalian cells. The erythroleukemia K562 cell line can endogenously express kindlin-3 and integrin α5β1, and we have shown previously that silencing kindlin-3 expression in K562 cells diminished integrin α5β1-mediated binding to fibronectin [26]. K562 cells also do not express kindlin-1 and kindlin-2, which are useful for interrogating the functions solely attributed by kindlin-3. In order to exclude any contribution from endogenous kindlin-3, CRISPR-Cas9n kindlin-3 gene knockout (KO) was performed, and the expression of kindlin-3 was verified by immunoblotting (Fig 6A). Next, kindlin-3 KO K562 cells were transfected and complemented with either WT (K3 WT) or AFA mutant (K3 AFA) kindlin-3 (Fig 6A). K562 cells expressed primarily β1 integrins. Based on flow cytometry analysis using a β1 integrin-specific monoclonal antibody (mAb), we showed that there was no marked difference in β1 integrin expression between K562 (K3 WT) and K562 (K3 AFA) cells (Fig 6B). Hence, any difference in β1 integrin-mediated cell adhesion and spreading between these cells is not attributed to a difference in β1 integrin expression. We then performed shear flow ligand-binding assay to determine the adhesive properties of these cells on fibronectin. As expected, kindlin-3 KO cells adhered less onto fibronectin compared with WT K562 cells even in the presence of exogenous activating agent $MnCl_2$. Notably, K3 AFA cells adhered significantly higher than that of K3 WT cells with or without $MnCl_2$, suggesting overt β1 integrin activation (Fig 6C). In line with this observation, K3 AFA cells also exhibited increased cell spreading on fibronectin compared with K3 WT cells (Fig 6D). Taken together, these results suggest that kindlin-3 exists as both monomers and trimers in mammalian cells, and that the latter represents an auto-inhibited state. Shifting the equilibrium between these 2 populations of kindlin-3 may be an important regulatory step in kindlin-3 function.

## Discussion

Kindlins are crucial regulators of integrin activation and outside-in signaling. Although kindlins have been shown to bind integrin β cytoplasmic tails, there is limited information on its regulation. Phosphorylation of kindlin-3 at Thr482/Ser484 has been shown to be important for kindlin-3-induced integrin activation, but the mode of action remains unclear [27]. Talin adopts a default low-affinity state for integrin-binding in which its "rod" region folds back and masks the integrin-binding pocket (also known as the PTB) in the F3 subdomain of its head region [28]. Conceivably, talin in an auto-inhibited state is important for preventing the deleterious activation of integrins. Unlike talins, kindlins do not possess a "rod" region. The structure of mouse PH-deleted kindlin-2 reveals a cloverleaf-like conformation in which the

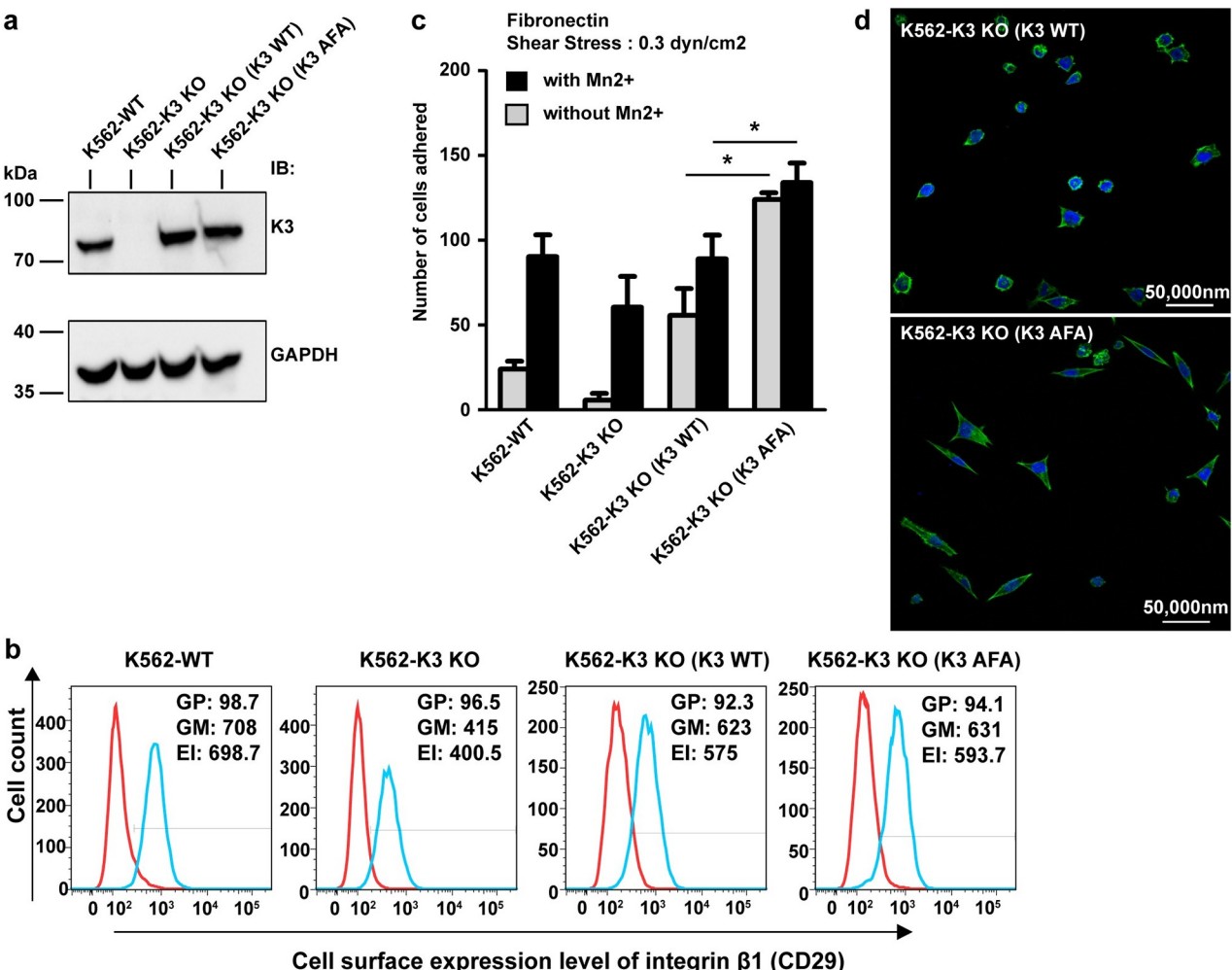

**Fig 6. Disruption of kindlin-3 trimerization exhibits overt integrin activation and increased cell spreading.** (a) Western blot analysis of kindlin-3 expression in WT K562 cells (K562-WT), kindlin-3 CRISPR-Cas9n KO cells (K562-K3 KO), and in the cells reconstituted with WT kindlin-3 (K3 WT) and AFA mutant kindlin-3 (K3 AFA). GAPDH was used as loading control. (b) Flow cytometry analysis of integrin β1 expression in aforementioned cells. Red histogram: control IgG. Blue histogram: anti-integrin β1 antibody. (c) Shear flow fibronectin-binding assay at a shear stress of 0.3 dyn/cm$^2$. Mn$^{2+}$: 0.5 mM MnCl$_2$. Each data point is the mean ± standard deviation from at least 3 independent experiments. Two-tailed unpaired *t* test was performed. $^*p < 0.05$. (d) Fluorescence microscope images of cells on fibronectin. Green: F-actin staining with Alexa Fluor 488 phalloidin. Blue: nuclei staining with DAPI. Scale bar: 50 μM. Additional information on numerical values and data was provided in S2 Data. AFA, triple mutations Q471A, A475F, and S478A; EI, expression index; F-actin, filamentous actin; GAPDH, glyceraldehyde 3-phosphate dehydrogenase; GM, geo-mean fluorescence; GP, percent gated positive; IgG, immunoglobulin G; KO, knockout; WT, wild type.

integrin-binding site in the F3 subdomain is exposed [18]. Hence, it is unlikely that kindlins can adopt an auto-inhibited state, if it exists, via intramolecular interactions. However, it does not rule out the possibility of auto-inhibition via homo-oligomerization.

In this study, we solved the structure of human full-length kindlin-3 (Fig 1B). We also showed that human kindlin-3 can form trimer that is incapable of binding the integrin β1A cytoplasmic tail (Figs 2–5 and S2–S10 Figs), suggesting that kindlin-3 can adopt an auto-inhibited state via homo-oligomerization. This could be explained by the observation that the integrin-binding pocket (PTB) in the F3 subdomain of one protomer is occluded by the last helix α2$_{PH}$ of another protomer in the structure of the kindlin-3 trimer. Furthermore, functional investigations of the AFA mutant kindlin-3 in cells through shear flow fibronectin-binding

assay and fluorescence microscope imaging are also in support of an auto-inhibited state of kindlin-3 homotrimer (Fig 6C and 6D). Our data appear contrary to a study in which mouse PH-deleted kindlin-2 forms a dimer and promotes integrin clustering [18]. Based on our data, the presence of the PH domain may interfere to a certain degree with dimer formation (Fig 3A and 3B). In line, the presence of full-length kindlin-3 as well as kindlin-2 dimers were not detected in our assays. However, we do not completely rule out the possibility of kindlins forming dimers under certain conditions given that the PH domain is linked to the F2 via flexible loops, allowing movement of the PH domain to possibly accommodate dimer formation. Along this line, we have provided evidence of kindlin-3 and kindlin-2 being able to form trimer, but the conditions leading to the formation or dissociation remains unknown. Taken together, these can be interesting future studies to pursue.

Kindlin-3 has been shown to promote integrin clustering [27,29]. Our findings that kindlin-3 trimer does not bind integrin β1A cytoplasmic tail suggest alternative mechanism(s) that need not involve kindlin oligomerization. First, integrin clustering can be induced by kindlins via their interactions with filamentous actin (F-actin) and focal adhesion proteins, including integrin-linked kinase (ILK) and paxillin [30–34]. Second, kindlins assist binding of talin to integrin β cytoplasmic tails. Hence, integrin clustering can be promoted by talin, which is known to form a dimer and bind F-actin [35]. Third, phosphoinositides are dynamically distributed in cells. Phosphatidylinositol(3,4,5)-trisphosphate (PtdIns(3,4,5)P3) is concentrated at the leading edge of neutrophils and *Dictyostelium discoideum* [36]. Hence, it is possible that the clustering of integrins by kindlins may be induced in part by the binding of kindlins to PtdIns(3,4,5)P3.

In conclusion, we resolved the structure of human full-length kindlin-3. Our data support the presence of kindlin-3 homotrimer, although the precise conditions (such as phosphorylation) leading to its formation or dissociation remain to be determined. Our data also suggest a novel mechanism by which the integrin-binding properties of kindlins can be inhibited by trimer formation. In addition to its role in the disease LADIII [37], kindlin-3 plays a crucial role in cancer progression [26,38,39]. Whether there is a link between oligomer formation of kindlin-3 or other kindlins and cancer progression will be of interest in future studies.

## Materials and methods

### Protein expression and purification

Using the human full-length kindlin-3 cDNA as template, the target gene was amplified by PCR and inserted into pET-24a expression vector (Novagen) between *NdeI* and *XhoI* restriction sites, introducing a C-terminal uncleavable 6xHis tag. The plasmid was transformed into *E. coli* BL21 (DE3) strain chemo-competent cells for protein expression.

The cells were grown at 37 ˚C in 2xYT medium (containing 30 μg/mL kanamycin) and cooled down to 16 ˚C once $OD_{600}$ reached approximately 0.8. Protein expression was induced with 0.5 mM isopropyl β-D-1-thiogalactopyranoside (IPTG) for 20 hours. Cells were collected by centrifugation at 4,000 *g* for 30 minutes and resuspended in lysis buffer: 50 mM $Na_2HPO_4$ (pH 8.5), 500 mM NaCl, 10% (v/v) glycerol, 10 mM imidazole, 5 mM mercaptoethanol (β-ME), and protease inhibitor (Thermo Fisher Scientific). Microfluidizer High Pressure Homogenizer LM20 was used to lyse the cell using pressure of 25,000 psi. Cell lysate was clarified by centrifugation at 40,000 rpm (Beckman Ti 45 rotor) for 1.5 hours, and the supernatant was used for protein purification. The target protein was sequentially purified by Ni-affinity (HisTrap HP column, GE Healthcare), ion-exchange (HiTrap Q HP column, GE Healthcare), and size-exclusion chromatography (Hiload16/60 Superdex 200 column, GE Healthcare) using buffer: 20

mM Tris-HCl (pH 7.5), 150 mM NaCl, and 5 mM β-ME. The purified protein was concentrated and frozen in liquid nitrogen.

The selenomethionine (SeMet)-labeled kindlin-3 was expressed in *E. coli* BL21 grown in M9 medium. Amino acid mixture (0.1 ng/mL L-lysine, 0.1 ng/mL L-phenylalanine, 0.1 ng/mL L-threonine, 0.05 ng/mL L-isoleucine, 0.05 ng/mL L-leucine, 0.05 ng/mL L-valine, and 0.025 ng/mL SeMet) was added before induction by IPTG, and the labeled protein was purified in the same manner as the native protein.

Similar protein purification protocols were used for native and mutant kindlin-3 expressed in insect cell expression system. The human full-length kindlin-3 gene was amplified by PCR and inserted into cloning vector pFB-LIC-Bse using ligation-independent cloning and introducing an N-terminal cleavable 6xHis tag. Baculovirus containing the target gene was amplified after 2 passages by transposition and transfection in insect cell Sf9. P2 virus was used to infect the Sf9 cell at 27 ˚C for 48 hours. Site-direct mutagenesis was performed to generate the mutant kindlin-3 (Q471A, A475F, and S478A) protein. The human full-length kindlin-2 was also expressed in insect cells and purified in the same manner.

Human ITGB1 (integrin beta tail residues 773–798) cDNA was amplified and inserted into pNH-TrxT vector with a N-terminal cleavable 6xHis thioredoxin tag by ligation-independent cloning. The plasmid was transformed into *E. coli* BL21 (DE3) Rosetta T1R cells for protein expression. The cells were grown at 37 ˚C in 2xYT medium with 30 μg/mL kanamycin and 34 μg/mL chloramphenicol, then cooled down to 16 ˚C once $OD_{600}$ was approximately 0.8, and protein expression was induced with 0.5 mM IPTG for 20 hours. Cells were collected by centrifugation at 4,000*g* for 30 minutes and resuspended in lysis buffer (20 mM HEPES [pH 7.5], 500 mM NaCl, 10 mM imidazole, 10% [v/v] glycerol, and 0.5 mM TCEP). Cells were lysed using LM20 (psi 25,000), and the cell lysate was centrifuged at 40,000 rpm (Beckman Ti 45rotor) for 1.5 hours, and the supernatant was used for purification. The target protein was purified by Ni-affinity (HisTrap HP column, GE Healthcare) and SEC (Hiload16/60 Superdex 75 column, GE Healthcare) using buffer: 20 mM HEPES (pH 7.5), 300 mM NaCl, 10% (v/v) glycerol, and 2 mM TCEP.

## Crystallization and data collection

Both native and SeMet-labeled kindlin-3 protein expressed in *E. coli* were concentrated to 9 mg/mL for crystallization trials. To facilitate kindlin-3 crystallization, surface residues mutagenesis with cluster 1 (E153A, K154A, and E155A) and cluster 2 (K513A, K515A, and K517A) was introduced based on Surface Entropy Reduction prediction (SERp) server. Hanging-drop vapor-diffusion method was used for crystallization. Crystals of native kindlin-3 were obtained under conditions containing 3 M sodium formate (pH 7.0), 3% w/v xylitol, and 9% (v/v) glycerol at 13 ˚C by seeding. SeMet-labeled kindlin-3 crystals were obtained under conditions containing 3 M sodium formate (pH 7.0), 3% (w/v) D-sorbitol, and 6% (v/v) glycerol at 20 ˚C, or 3 M sodium formate (pH 7.0) and 3% (w/v) D-(+)-trehalose dihydrate at 20 ˚C by seeding. Seeding was done by transferring 1 crystal into 100-μL reservoir, crushing it down to nuclei, and adding them into the drop. Crystals were flash frozen in liquid nitrogen for data collection at National Synchrotron Radiation Research Center (Taiwan) and Swiss Light Source.

Diffraction data from SeMet-labeled and native crystals of kindlin-3 were collected at X06DA-PXIII beamline in the Swiss Light Source (SLS, Switzerland), Switzerland. SAD measurement on SeMet kindlin crystals were performed using a multi-orientation high-multiplicity data collection strategy [40]. Each scan of 360 was collected at different orientation of the crystal with 0.2˚ of oscillation angle, 0.1-second exposure time on Pilatus 2MF detector at X-ray energy of 12.67 keV (i.e., Se-edge). Twelve phi-scans of 1,800 frames each were collected at

different positions of a single crystal. For the native kindlin-3 crystal, datasets were collected at 1.0 Å with oscillation angle 0.200˚. Five phi-scans of 1,000 frames each were collected at different positions of a single crystal.

## Data processing and structure determination

The data were processed with XDS package [41], and structures were solved with Phenix program [20]. The diffraction patterns of each native dataset were individually processed using XDS and merged using XSCALE with Friedel's Law set to TRUE. The merged data were then converted to MTZ format using XDSCONV and F2MTZ. The SeMet datasets were processed, merged, and converted similarly to the native datasets, with the exception that the Friendel's Law was set to FALSE. The initial search of the heavy atom sites was performed with SHELX combined with phenix.autosol, yielding 30 sites with reasonable occupancy in an asymmetric unit (11 non-START methionine theoretically exists in each kindlin-3 molecule) [19,20]. The phasing statistics are listed in Table 1:

Initial model building was performed using phenix.autobuild, and 3 copies of the molecule could be recognized from the segments built. Combining with the number of the heavy atom sites, a trimer was assigned to an asymmetric unit. The structure of kindlin-2 from *Mus musculus* (PDB entry: 5XPZ) was superimposed onto the built segments, and refinement of the fitted trimer was performed using phenix.refine, which revealed empty density near the F1 subdomain. Because the model structure (PDB entry: 5XPZ) lacked the PH domain, we search the map for a PH domain homologue (PDB entry: 5L81) using molecular replacement with phenix.phaser. The model was then refined iteratively using phenix.refine. The refined model was used to perform molecular replacement (phenix.phaser) with the native data. The model sequence was then corrected using Chainsaw, and the model was refined iteratively using phenix.refine and Lorestr [42]. Note that for the native data, we collected 3 highly redundant datasets from 3 isomorphous crystals and merged the data together. The data collection and refinement statistics are summarized in S1 Table. All the structure figures were rendered in PyMol (Schrödinger, LLC).

## Analytical gel filtration chromatography

Analytical gel filtration chromatography was carried out with AKTA Explore system (GE Healthcare). Superdex 200 Increase 10/300 GL column (GE Healthcare) pre-equilibrated with the buffer 20 mM Tris-HCl (pH 7.5), and 150 mM NaCl was used for all protein samples. Gel filtration calibration kit HMW (GE Healthcare) containing 5 well-defined proteins with molecular weight ranging from 43k to 669k Da, and Blue Dextran 2000 was used to generate a standard calibration curve.

## Negative staining EM imaging and processing

The target protein was diluted to 0.01 mg/mL after gel filtration for negative staining EM. Carbon-coated TEM grid (Electron microscopy sciences, 200 mesh Copper) was used. The grid was glow discharged for 35 seconds just before use, and 4 μl of the sample was loaded on to the

**Table 1. Phasing statistics.**

| Phasing resolution | Space group | Se Sites | Anomalous Completeness | FOM |
|---|---|---|---|---|
| 49.59–3.80 | C 2221 | 31 | 99.81% | 0.25 |

grid and incubated for 1 minute before absorbing the solution. Next, 4 µl of 2% uranyl acetate was then slowly pipetted on to grid to cover the protein sample for 1 minute and absorbed. The gird was loaded to a Tecnai T12 (FEI) electron microscope operated at 120 kV. Digital micrographs were recorded at 43,000× magnification with defocus range between −1 and −2 µm using an Eagle 4K x 4K camera. EMAN2 was used for particle visualization [43].

## Crosslinking and MS analysis

DSSO was purchased from Thermo Fisher Scientific and used for crosslinker. First, 1 mg DSSO was dissolved into 100 µL DMSO to prepare a 25 mM stock solution before use. Both monomeric and trimeric kindlin-3 purified from insect cells were applied for DSSO treatment. Protein was dissolved in 40 µL of 20 mM HEPES buffer (pH7.5) at 0.2 µM. Next, 0.1 µL of DSSO stock solution was added to the protein sample, and the final reaction buffer was adjusted to 45 µL with a 300-fold molar excess of crosslinker over the protein concentration. A control sample containing the protein without crosslinker was also prepared. Both samples were incubated at room temperature for 1 hour, and the reaction was quenched by adding 1M Tris buffer (pH 8.5) to a final concentration of 20 mM. The crosslinked protein was analyzed by SDS-PAGE and MS.

The 2 bands corresponding to kindlin-3 trimers from 2 crosslinking experiments were excised from the SDS-PAGE gel for in-gel digestion. After reduction using DTT, alkylation with IAA, and dehydration by ACN, the protein in the gel matrix was first digested by lys-C for 4 hours followed with overnight trypsin digestion. The peptides were resuspended in 0.1% formic acid and analyzed on a Dionex Ultimate 3000 RSLCnano system coupled to a Q-Exactive tandem mass spectrometer (Thermo Fisher). The LC-MS/MS was operated at an electrospray potential of 1.5 kV. A full MS scan (350–1,600-m/z range) was acquired at a resolution of 70,000 at m/z 200 and a maximum ion accumulation time of 100 milliseconds. Peptide ions with charge state equal or less than 3+ were excluded from MS/MS. The LC-MS/MS raw data files were searched for crosslinked peptides using pLINK2 [44]. The search values included trypsin digestion, DSSO cleavable crosslinker, 10 ppm precursor mass tolerance, and 0.02 Da fragment mass tolerance. We have done 2 replicates, with those consistently observed crosslinked peptides in all experiments/replicates shown in the S8C Fig and S1 Data.

The MS crosslinking proteomics data have been deposited to the ProteomeXchange Consortium via the PRIDE partner repository with the dataset identifier PXD019110 (ftp://ftp.pride.ebi.ac.uk/pride/data/archive/2020/06/PXD019110).

## CD spectroscopy measurement

CD experiments were carried out on a calibrated CD spectrometer (Chirascan, Applied Photophysics) at 25 ˚C in the far UV region (190–260 nm wavelength). Purified protein samples were dissolved in 50 mM sodium phosphate buffer (pH 7.5) at a concentration of 0.1 mg/mL. Next, 200 µL protein solution was loaded into a 1-mm path length quartz glass cuvette (Hellma Analytics) for measurement. The scan rate was set at 0.5 second per point with an approximate 65 seconds scanning in total. Data were analyzed by CDNN software [45].

## ITC assay

ITC measurements were carried out on a MicroCal PEAQ-ITC (Malvern) at 25 ˚C. All protein samples were dissolved in buffer: 50 mM HEPES (pH 7.5) and 150 mM NaCl. The titration processes were performed by injecting 2 µl aliquots of the ITGB1 tail, or for control His-Trx tag (no target protein) at a concentration of 500 µM into monomer or trimer

kindlins at a concentration of 50 μM with a stirring speed of 600 rpm. The titration data were analyzed using the MicroCal PEAQ-ITC Analysis Software.

## SEC-MALS

The SEC-MALS system was comprised of HPLC system coupled with UV and fluorescence detector (Shimadzu), Dawn Heleos II Multi-Angle Light Scattering detector (Wyatt) and Opti-lab T-rEX refractive index (RI) detector (Wyatt). Superdex 200 increase 5/150 GL (GE) was used to separate the species in solution. The system was pre-equilibrated with buffer containing 20 mM HEPES (pH 7.5), 500 mM NaCl and the RI detector was purged for 2 column volumes until the dRI value was stable. The dRI value was used for data analysis later. Protein samples were dissolved in 50 μL of the same buffer with at least 2 mg/mL concentration and filtered through 0.1-μm filter before injection. Astra software was used for sample auto-injection and data analysis.

## Generation of stable HEK 293 cell line expressing His$_6$-tagged kindlin-3

6xHis-3xGly(linker)-kindlin-3 (WT) in pcDNA3.1-zeo(-) plasmid was generated by PCR using the HA-kindlin-3 pcDNA3.1 expression plasmid[29] as a template and the following primers:

5'CTAGCTAGCATGCACCACCACCACCACCACGGTGGTGGTATGGCGGGGGATGAAG ACAG3' and 5'CCCAAGCTTTCAGAAGGCCTCATGGCCCCCGGTGAGCTGCAGGA AGAGGTCTTC3'.

The PCR product was digested with restriction enzymes NheI and HindIII and ligated into pcDNA3.1-zeo(-) digested with the same enzymes. HEK 293 (American Type Culture Collection [ATCC], Manassas, VA) cells were cultured in DMEM media supplemented with 10% HI-FBS and Penicillin/Streptomycin. Cells ($1 \times 10^6$) were seeded into one well of a 6-well plate the day before transfection. Cells were transfected with 3 μg of plasmid by PEI followed by selection in Zeocin (Invitrogen, Waltham, MA) at 200 μg/mL for 2 weeks to establish a stable cell line. The expression of kindlin-3 was verified by western blotting. For purification of kindlin-3, 5 batches of $12 \times 10$ cm culture dish (approximately 80% cell confluence per dish) of cells were harvested. Purification steps and conditions were the same as for *E. coli* and insect cell expression.

## Bead-based BiFc analysis of kindlin-3 homotrimer

The expression plasmids N-eYFP (M1-A155)-12Gly (linker)-kindlin-3 (WT or AFA mutant) and C-eYFP (D156-K239)-12Gly (linker)-kindlin-3 (WT or AFA mutant) were generated by Gibson assembly. The N-eYFP fragment containing part of the 12 Gly linker was generated by PCR using the peYFP-N1 plasmid (Clontech) as the template and the following primers: 5'TATAGGGAGACCCAAGCTGGATGGTGAGCAAGGGCGAGGAC3' and 5'CACCGC CTCCTCCGCCACCTCCGCCACCGGCCATGATATAGACGTTGTGGCTG3'.

The C-eYFP fragment containing part of the 12 Gly linker was similarly generated but using different primers:

5'TATAGGGAGACCCAAGCTGGATGGACAAGCAGAAGAACGGCATCAAGG3' and 5'CACCGCCTCCTCCGCCACCTCCGCCACCCTTGTACAGCTCGTCCATGCCG3'. Kindlin-3 fragment (WT or AFA mutant) containing part of the 12 Gly linker at its N-terminus was generated by PCR using either kindlin-3 WT or kindlin-3 AFA plasmid as the template and the following primers:

5'AGGTGGCGGAGGAGGCGGTGGAGGCGGTATGGCGGGGATGAAGACAGCC3' and
5'GGCTGATCAGCGGTTTAAACTCAGAAGGCCTCATGGCCCC3'.

The pcDNA3.1-zeo(-) vector was linearized by restriction enzymes NheI and AlfII. N-eYFP or C-eYFP fragment, kindlin-3 (WT or AFA mutant) fragment, and linearized pcDNA3.1-zeo(-) plasmid were mixed in Gibson assembly master mix (New England Biolabs, Ipswich, MA) and incubated at 50 °C for 50 minutes. The assembled plasmid was transformed into *E. coli* competent cells for plasmid amplification. The HA-kindlin-3 pcDNA3.1 expression plasmid has been reported previously [29]. To generate F3 deletion (F3Δ), site-directed mutagenesis was performed on N-eYFP (M1-A155)-12Gly (linker)-kindlin-3 WT, C-eYFP (D156-K239)-12Gly (linker)-kindlin-3 WT, and HA-kindlin-3 plasmids using the primers:

5'AGTCCCTGCCCGACTTCTGAATCTCCTATGTCATGG3' and 5'CCATGACATAGGAG
ATTCAGAAGTCGGGCAGGGACT3' such that a stop codon was introduced at Gly554.

HEK 293T (ATCC) cells ($3\times10^6$) were cultured in DMEM media supplemented with 10% HI-FBS and penicillin/streptomycin. Cells were cotransfected using polyethylenimine (PEI) with equal amount (ratio 1:1:1) of HA-kindlin-3, N-eYFP (M1-A155)-12Gly (linker)-kindlin-3, and C-eYFP (D156-K239)-12Gly (linker)-kindlin-3 plasmids. The 3 plasmids transfected were WT, AFA, and F3Δ kindlin-3. Next day upon transfection, cells were lysed in lysis buffer (1% NP-40, 150 mM NaCl, and 10 mM Tris [pH 7.4]) containing protease inhibitor (Sigma-Aldrich). A small fraction of the cell lysate was analyzed by western blotting using rat anti-kindlin-3 antibody (1:1,000 dilution) (clone 181A, generated in-house). The remaining cell lysate was separated into 2 equal volumes. One half of the cell lysate was mixed with 3 μg of mouse anti-HA (Sigma-Aldrich) antibody and the other half with 3 μg of mouse serum IgG (Sigma-Aldrich). Thereafter, 40 μg of SureBeads Protein G Magnetic beads (Bio-Rad, Hercules, CA) were added to each sample and incubated at 4 °C for 30 minutes on a roller. Beads were washed in lysis buffer 3 times and resuspended in 300 μL of ice-cold PBS followed by the detection of YFP signal detection on a BD 3-Lasers Fortessa X20 flow cytometer with High-Throughput Sampler (HTS) Becton Dickinson, Franklin Lakes, NJ. FlowJo software (Tree Star, Ashland, OR) was used to analyze the data.

## Generation of kindlin-3 KO K562 cells and reconstitution of kindlin-3 expression

The human erythroleukemia cell line K562 was obtained from the ATCC and maintained in complete RPMI medium containing 10% heat-inactivated fetal bovine serum, 100 IU/mL of penicillin, and 100 μg/mL of streptomycin. Kindlin-3 KO K562 cells were generated by the CRISPR-Cas9n method. The 2 gRNA targeting sequences 5'-GGGCTACCGCCAACACT GGG-3' and 5'-GGGGCCTTCGTGGGATGCTG-3' were each subcloned into the pSpCas9n (BB)-2A-GFP (PX461) plasmid (a gift from Feng Zhang, Broad Institute of MIT and Harvard, Addgene plasmid #48140) [46]. K562 cells were cotransfected with both plasmids by electroporation. Transfected GFP-fluorescent cells were sorted on a FACS-Aria cell sorter (Becton Dickinson, Franklin Lakes, NJ) and subcloned (1 cell/well) in 96 well plates. After culture expansion, clones were screened for the complete loss of kindlin-3 expression by western blotting using the anti-kindlin-3 mAb clone 9 [29] and verified by genomic DNA sequencing of the CRISPR-Cas9n targeted region in the kindlin-3 gene. A K562 clone with complete KO of kindlin-3 expression was used in the following reconstitution experiments. Cells were transfected by electroporation with expression plasmid pcDNA3.1(zeo) containing cDNA of either WT kindlin-3 or kindlin-3 Q471A/A475F/S478A (henceforth referred to as AFA mutant)

followed by selection in 100 µg/mL Zeocin to generate stable lines. The expression of WT kindlin-3 and kindlin-3 AFA mutant in these cells was verified by western blotting.

### Flow cytometry analysis

Surface expression of β1 integrins on K562 cells was determined by staining cells with anti-CD29 (β1) antibody (Becton Dickinson-Pharmingen, Franklin Lakes, NJ) or control IgG from rat serum (Sigma Life-Science) and FITC-conjugated anti-Rat IgG (Sigma Life-Science) followed by analysis on a FACSCalibur flow cytometer (Becton Dickinson, Franklin Lakes, NJ). Data were analyzed and presented using the FlowJo software (Tree Star, Ashland, OR).

### Shear flow assay

Shear flow assay was performed as previously described [26] with slight modifications. Briefly, µ-slide I$^{0.4}$ Luer parallel flow chamber (Ibidi GmbH, Germany) was coated with 5 µg/cm$^2$ of human fibronectin (Sigma Gibco) in PBS overnight at 4 ˚C followed by blocking of nonspecific binding sites in PBS containing 0.5% (w/v) BSA. The flow chamber was assembled on a microscope stage housed in a 37 ˚C incubator. Cells were resuspended in Hanks' Balanced Salt Solution without or with 0.5 mM MnCl$_2$. Cells were infused into the chamber at a shear stress of 0.3 dyn/cm$^2$ shear using a syringe pump. Total number of adherent cells in 4 different fields of the chamber were counted. The average number of cells per field was then calculated.

### Fluorescence staining and imaging

Cells were resuspended in culture media containing 0.5 mM MnCl$_2$ and seeded on a coverslip glass-bottom tissue culture dish (MatTek, Ashland, MA) that was precoated with human fibronectin (Gibco) at concentration of 10 µg/mL. Cells were allowed to adhere on fibronectin under culture conditions for 2 hours. Medium was discarded, and cells were fixed in 3.7% (w/v) paraformaldehyde-PBS solution at RT for 10 minutes. Cells were then incubated in CSK buffer (100 mM NaCl, 300 mM sucrose, 3 mM MgCl$_2$, 1 mM EGTA, 10 mM PIPES [pH 6.8]) containing 0.3% (v/v) Triton at RT for 1 minute. Cells were washed 3 times with PBS and nonspecific sites blocked with PBS containing 5% (w/v) BSA overnight at 4 ˚C. Fixed cells were stained with DAPI and Alexa Fluor 488 phalloidin (Invitrogen Corporation, Carlsbad, CA). Cells were visualized under confocal laser scanning microscope LSM710 (Zeiss, Oberkochen, Germany) at a magnification of 40× and 100× under oil immersion.

## Supporting information

**S1 Fig. Structure-based sequence alignment of human kindlins.** The domains and the F1 loop are highlighted in the same manner as in Fig 1A. Secondary structure motifs and numbering are based on kindlin-3.
(TIF)

**S2 Fig. The anomalous difference Fourier map for total Se atom positions.** (**a**) The anomalous difference Fourier map is generated using Global Phasing Buster. The map is displayed with Se sites as green spheres and contoured at 3 sigma. Total 31 Se sites were shown in the unit cell. (**b**) The anomalous difference Fourier map (white mesh) is displayed with the final refined model (semitransparent cartoon rendering). The sulfur atoms on methionines of the final model are shown as green spheres. A zoom-in view of the methionine sites is also presented with 3 methionine side chains shown in stick model. Se, selenine.
(TIF)

**S3 Fig. Representative 2Fo-Fc electron maps.** The representative 2Fo-Fc electron density maps are shown in blue meshes with the ribbon model of the protein. Secondary structure elements are labeled. (**a**) 2Fo-Fc electron density map of protomer–protomer interface. (**b**) 2Fo-Fc electron density map of F2 subdomain of one protomer. Fc, calculated structure factor; Fo, observed structure factor.
(TIF)

**S4 Fig. Crystal packing of C-alpha models with unit cell dimensions.** Each trimer is colored identically.
(TIF)

**S5 Fig. Fo-Fc electron density map of the loop between α2$_{F2}$ and β1$_{PH}$.** The modeled loop between α2$_{F2}$ and β1$_{PH}$ appears to be a helix interacting with α2$_{PH}$ to stabilize the entire domain. Fc, calculated structure factor; Fo, observed structure factor; PH, pleckstrin homology.
(TIF)

**S6 Fig. Negative staining electron microscopy of kindlin-3 trimer.** (**a**) Typical negative stain electron microscopy micrograph of kindlin-3 trimer purified from Sf9 cells. Kindlin-3 particles are highlighted by white squares. (**b**) Close-up view of kindlin-3 particles. Sf9, *Spodoptera frugiperda* 9.
(TIF)

**S7 Fig. DSSO crosslinked kindlin-3.** (**a**) SDS-PAGE of kindlin-3 monomer with or without DSSO treatment. Lane 1 indicates the native kindlin-3 monomer purified from insect cells. Monomeric kindlin-3 in solution gave a band above 70k Da. Lane 2 indicates the kindlin-3 monomer crosslinked by DSSO. Crosslinked trimeric kindlin-3 in solution exhibited a band above 200k Da (labeled by red arrow). (**b**) Analytical gel filtration chromatography profiles of kindlin-3 monomer with or without DSSO treatment. K3F monomer without DSSO treatment (blue) and K3F monomer with DSSO treatment (red): K3F monomer without DSSO treatment only exhibits monomeric state, whereas K3F monomer with DSSO treatment exhibits both monomeric and trimeric states. Note that molecular weight markers for analytical gel filtration chromatography are indicated by black arrows. (**c**) SDS-PAGE of kindlin-3 trimers with or without DSSO treatment. Lane 1 indicates the native kindlin-3 trimer purified from insect cells. Trimeric kindlin-3 in solution was denatured into monomeric state to give a band above 70k Da. Lane 2 indicates the kindlin-3 trimer crosslinked by DSSO. Crosslinked trimeric kindlin-3 in solution exhibited a band above 200k Da (labeled by red arrow). DSSO, disuccinimidyl sulfoxide.
(TIF)

**S8 Fig. DSSO crosslinked residue pairs detected by MS.** (**a**) Lysine–lysine intra- (red) and inter- (blue) molecules crosslinks were mapped onto the kindlin-3 crystal structure. The intermolecular crosslink marked with a blue asterisk is approximately 30 Å. It was identified with very high confidence. The intermolecular crosslink marked with a green asterisk is approximately 34 Å. It was identified with a relatively low confidence but also appears reasonable upon inspection of the structure. Besides, both 2 intramolecular crosslinks were identified with very high confidence. (**b**) Two domain organization of kindlins showing the identified lysine–lysine crosslinks. K567-K589 and K262-K457 are intramolecular crosslinks. K457-K567 and K252-K457 are intermolecular crosslinks. (**c**) Annotated MS/MS spectrum showing the b and y fragment ions of intermolecular crosslinked peptides K(252)DEILGIANNR-LASK(457)GR. DSSO, disuccinimidyl sulfoxide; MS, mass spectrometry.
(TIF)

**S9 Fig. Circular dichroism spectra of kindlin-3 monomer from *E. coli* and Sf9 cells.** The far UV spectrum shows that *E. coli*–expressed kindlin-3 and Sf9-expressed kindlin-3 have very similar secondary and tertiary structures. *E. coli*, *Escherichia coli*; Sf9, *Spodoptera frugiperda* 9. (TIF)

**S10 Fig. Binding assay of integrin β1 tail and human full-length kindlins using ITC.** Note that in the individual figure, the upper panel shows binding isotherm, and the lower panel shows data-fitting curve. (**a**) Binding assay for kindlin-3 Sf9 monomer. The protein tested is the monomer form of native kindlin-3, which is expressed in Sf9 insect cells. ITC measurement demonstrated a moderate binding between integrin β1 tail and monomeric kindlin-3. (**b**) Binding assay for kindlin-3 Sf9 trimer. The protein used is native kindlin-3 trimer, which is expressed in Sf9 insect cells. In agreement with our structural data (Fig 3C), kindlin-3 trimer shows no binding to integrin β1 tail. (**c**) Binding assay for kindlin-2 Sf9 monomer. The protein used is native kindlin-2 monomer expressed in Sf9. Compared with monomeric kindlin-3, ITC measurement indicated a much stronger binding between integrin β1 tail and monomeric kindlin-2. (**d**) Binding assay for kindlin-2 Sf9 trimer. The protein used is native kindlin-2 trimer expressed in Sf9. In agreement with kindlin-3 trimer, kindlin-2 trimer shows no binding to integrin β1 tail. ITC, isothermal titration calorimetry; Sf9, *Spodoptera frugiperda* 9. (TIF)

**S11 Fig. Structural comparison of F2 domains.** The F2 domains are from kindlin-3, kindlin-2, and talin, colored deep teal, yellow, gray, respectively. (TIF)

**S1 Data. Annotated MS/MS spectra of DSSO crosslinked peptides that are identified with pLink2 software.** DSSO, disuccinimidyl sulfoxide; MS, mass spectrometry. (ZIP)

**S2 Data. Additional information on numerical values and data presented in Figs 5C, 6B and 6C.** (ZIP)

**S1 Table. Data collection and refinement statistics.** (DOCX)

## Acknowledgments

We thank C. W. Liew for his help with initial crystal screening in-house and A. Wong for his guidance on negative screening sample preparation.

## Author Contributions

**Conceptualization:** Wenting Bu, Suet-Mien Tan, Yong-Gui Gao.

**Data curation:** Wenting Bu.

**Formal analysis:** Wenting Bu, Suet-Mien Tan, Yong-Gui Gao.

**Funding acquisition:** Suet-Mien Tan, Yong-Gui Gao.

**Investigation:** Wenting Bu, Zarina Levitskaya, Zhi Yang Loh, Shengyang Jin, Shibom Basu, Rya Ero, Xinfu Yan, Meitian Wang, So Fong Cam Ngan, Siu Kwan Sze, Suet-Mien Tan, Yong-Gui Gao.

**Methodology:** Suet-Mien Tan, Yong-Gui Gao.

**Project administration:** Suet-Mien Tan, Yong-Gui Gao.

**Resources:** Suet-Mien Tan, Yong-Gui Gao.

**Supervision:** Suet-Mien Tan, Yong-Gui Gao.

**Validation:** Wenting Bu, Yong-Gui Gao.

**Visualization:** Yong-Gui Gao.

**Writing – original draft:** Wenting Bu, Suet-Mien Tan, Yong-Gui Gao.

**Writing – review & editing:** Wenting Bu, Rya Ero, Suet-Mien Tan, Yong-Gui Gao.

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
