## [Editor Report · Decision Letter 0]

12 Aug 2019

Dear Dr Gao, 

Thank you for submitting your manuscript entitled "Structural basis of human full-length kindlin-3 homotrimer in an auto-inhibited state" for consideration as a Research Article by PLOS Biology.

Your manuscript has now been evaluated by the PLOS Biology editorial staff as well as by an academic editor with relevant expertise and I am writing to let you know that we would like to send your submission out for external peer review.

*Please be aware that, due to the voluntary nature of our reviewers and academic editors, manuscripts may be subject to delays during the holiday season. Thank you for your patience.*

Please re-submit your manuscript within two working days, i.e. by Aug 14 2019 11:59PM.

Kind regards,

Lauren A Richardson, Ph.D

Senior Editor

PLOS Biology

---

## [Decision Letter · Decision Letter 1]

29 Aug 2019

Dear Dr Gao,

Thank you very much for submitting your manuscript "Structural basis of human full-length kindlin-3 homotrimer in an auto-inhibited state" for consideration as a Research Article at PLOS Biology. Your manuscript has been evaluated by the PLOS Biology editors, an Academic Editor with relevant expertise, and by independent reviewers.

The reviews of your manuscript are appended below. You will see that the reviewers find the work potentially interesting. However, based on their specific comments and following discussion with the academic editor, I regret that we cannot accept the current version of the manuscript for publication. We remain interested in your study and we would be willing to consider resubmission of a comprehensively revised version that thoroughly addresses all the reviewers' comments. We cannot make any decision about publication until we have seen the revised manuscript and your response to the reviewers' comments. Your revised manuscript would be sent for further evaluation by the reviewers.

You will see that the reviewers find the work potentially interesting. However, based on their specific comments and following discussion with the academic editor, I regret that we cannot accept the current version of the manuscript for publication. We remain interested in your study and we would be willing to consider resubmission of a comprehensively revised version that thoroughly addresses all the reviewers' comments. We cannot make any decision about publication until we have seen the revised manuscript and your response to the reviewers' comments. Your revised manuscript would be sent for further evaluation by the reviewers.

Having discussed the reviews with the academic editor, we think that the data supporting the formation and function of the kindlin-3 trimer must be improved. The reviewers recommend a number of different assays and techniques that should be applied. Additionally, we agree with Reviewer #2 and believe that better in vivo support of trimer formation and auto-inhibition is needed. Lastly, the Academic Editor highlights that the quality of the X-ray structures should be improved or at the very least, the limitations of the resolution should be addressed.

We appreciate that these requests represent a great deal of extra work, and we are willing to relax our standard revision time to allow you six months to revise your manuscript. Please email us (plosbiology@plos.org) to discuss this if you have any questions or concerns, or think that you would need longer than this. At this stage, your manuscript remains formally under active consideration at our journal; please notify us by email if you do not wish to submit a revision and instead wish to pursue publication elsewhere, so that we may end consideration of the manuscript at PLOS Biology.

Your revisions should address the specific points made by each reviewer. Please submit a file detailing your responses to the editorial requests and a point-by-point response to all of the reviewers' comments that indicates the changes you have made to the manuscript. In addition to a clean copy of the manuscript, please upload a 'track-changes' version of your manuscript that specifies the edits made. This should be uploaded as a "Related" file type. You should also cite any additional relevant literature that has been published since the original submission and mention any additional citations in your response. 

Before you revise your manuscript, please review the following PLOS policy and formatting requirements checklist PDF: http://journals.plos.org/plosbiology/s/file?id=9411/plos-biology-formatting-checklist.pdf. It is helpful if you format your revision according to our requirements - should your paper subsequently be accepted, this will save time at the acceptance stage.

Please note that as a condition of publication PLOS' data policy (http://journals.plos.org/plosbiology/s/data-availability) requires that you make available all data used to draw the conclusions arrived at in your manuscript. If you have not already done so, you must include any data used in your manuscript either in appropriate repositories, within the body of the manuscript, or as supporting information (N.B. this includes any numerical values that were used to generate graphs, histograms etc.). For an example see here: http://www.plosbiology.org/article/info%3Adoi%2F10.1371%2Fjournal.pbio.1001908#s5.

For manuscripts submitted on or after 1st July 2019, we require the original, uncropped and minimally adjusted images supporting all blot and gel results reported in an article's figures or Supporting Information files. We will require these files before a manuscript can be accepted so please prepare them now, if you have not already uploaded them. Please carefully read our guidelines for how to prepare and upload this data: https://journals.plos.org/plosbiology/s/figures#loc-blot-and-gel-reporting-requirements.

Upon resubmission, the editors will assess your revision and if the editors and Academic Editor feel that the revised manuscript remains appropriate for the journal, we will send the manuscript for re-review. We aim to consult the same Academic Editor and reviewers for revised manuscripts but may consult others if needed.

If you still intend to submit a revised version of your manuscript, please go to https://www.editorialmanager.com/pbiology/ and log in as an Author. Click the link labelled 'Submissions Needing Revision' where you will find your submission record. 

Sincerely,

Lauren A Richardson, Ph.D

Senior Editor

PLOS Biology

Reviews

Reviewer #1: 

This is potentially significant manuscript showing that kindlin-3 might form homotrimer in an autoinhibited state to prevent integrin binding and activation. Previous structural studies on homologous kindlin-2 showed a crystal structure with deletion of PH domain and the protein undergoes monomer/dimer transition although the dimer formation was extremely slow (takes several days). The monomeric structure in this manuscript is convincing but I have major concerns in the trimer study that is the core of the manuscript. 

Major points:

1. The authors claimed that they observed pure monomer of kindlin-3 in E. Coli system but monomer/trimer equilibrium in insect cell system (sf9). However, the whole manuscript did not describe any experimental conditions about how they observed trimer by gel-filtration. The trimer claim in Fig 2d is not convincing at all since this is supdex 200 with no molecular marker indicated. I can argue that that peak is a dimer or an oligomer or aggregate. Oligomer or aggregate is actually possible since negative staining in Fig. 2f showed a 10 nm particle size that is unrealistically big for a 228 kD trimer. Much more convincing evidence for the trimerization is needed by some definitive techniques such as analytic ultracentrifugation and light scattering. 

2. The claimed trimer from insect cells contradicts from a previous biophysical study using the same system, which showed only monomer (ref 16). Therefore, the authors’ explanation that only insect cell system produces trimer is not correct. Is it possible that the authors observed some concentration-dependent oligomerization or aggregation? Again, no experimental conditions were provided for how they detected trimer vs monomer.

3. The ITC studies on the trimer vs monomer binding to integrin are not convincing since ITC is more reliable on strong binding Kd<10 microM). The monomer kindlin-3 binds to integrin at ~200 microM here, which is extremely weak and usually not detectable by ITC. Some alternative method is crucial for confirm the integrin binding studies by ITC. For example, how about GST pull down to show the difference? The authors claim the trimer formation is conserved among kindlins. Since kindlin-1/2 bind much tighter to integrins, which may be easier to be measured, the authors should use one of them (preparation of these kindlins is well established in literatures) to demonstrate this crucial point, i.e., monomer binds better to integrin than trimer. The experiment may provide evidence for their conclusion that the trimer based autoinhibition is conserved among kindlins. This is very important for this manuscript!

4. From the crystallographic site, the authors reported the crystal SG of C2221, where z equals 8, resulting in several options of monomer numbers per the asymmetric unit. Based on the reported crystal parameters, the authors determine three molecules in the asymmetric unit, resulting in quite outlier values of Matthews probability and solvent content (Vm=3.47 A**3/Da; Solvent=64.58%). Have the authors ever tested to analyze four molecules (as dimer of dimer) in the asymmetric unit, where more feasible values could be accountable (Vm=2.60 A**3/Da; Solvent=52.77%)? Have the authors encountered potential pseudo-symmetry problem? The current resolution (3.6A) in the manuscript is low, which may cause ambiguity of the trimer determination. In this regard, the Rmerge was not reported in the statistic table but is required for crystal structure report. The authors need to state the Rmerge or equivalent statistic value during data collection and processing (R-meas if XDS program utilized). If not clear, the entire log file of CORRECT.LP needs to be supplemented. As a rule of thumb, the cut-off of I/sI in the outer shell must be satisfied ≥ 2.0 unless specific reason being provided. CC(1/2) in the outer shell must be also ≥ 70%. If the above criteria can affect the high-resolution cut-off, the authors may need to correct the resolution limit followed by structure refinement. 

5. The detailed phasing statistics need to be provided in the method section. Anomalous difference Fourier map for total Se atom positions is also required in the supplementary figure. Representative electron density maps (2Fo-Fc) need to show in the supplementary figure. Crystal packing of C-alpha models with unit cell dimensions need to be presented in the supplementary figure. 

Minor points:

1. Line 105-106, please provide the technical details to convince readers about the expression, purification and sample preparation as to how the trimer vs monomer was obtained and crystallized.

2. Line 142-144, please discuss why authors ascertain which K3 molecule the PH domain belongs to after all the limited electron density connecting PH domain to F2 domain. 

3. Line 239-240, the authors proposed that S484 phosphorylated Kindlin-3 monomer is the ultimate activation state, however S484A K3 monomer mutant only exhibited 1M Kd on ITC. Please discuss how such weak interaction can be biologically significant. 

4. Typo on line 240, week � weak.

5. Line 244-246, residue S484 didn’t show in Fig 2C to support the sentence.

6. Line 248-249, it is confusing that S484 phosphorylation does not play the major role in kindlin-3 trimer formation, but somehow affect the kindlin-3 function. Does PH domain play the major role, such as PH domain recruitment to the membrane lipid?

7. Line 266-269, the statements here are unsubstantiated because i. the K3-AFA mutant interaction with integrin b1 has not been tested; ii. the authors need to reveal the technical details of how they obtained monomer and trimer K3 in sf9 cells, and confirm the K3-AFA is a monomer even expressed in the buffer condition for trimer.

8. Fig. 2d and S6, a molecular weight marker should be included.

9. Fig. 2e, please box a larger area to include more particles.

---------------

Reviewer #2: 

The submitted manuscript discusses the structure of full length kindlin-3. A new mechanism of kindlin-3 regulation is proposed, namely the inactivation of kindlin-3 (inability to bind integrins) via homotrimerization.

I would like to say that as a non-structural biologist I was able to enjoy the manuscript as it was very well written and easy to follow. The structural data appears sound although others are likely much more competent to comment. I found it surprising however that monomeric kindlin-3 from insect cells bound to beta1 integrin tails with a surprisingly high KD of 200 micromolar (and no interaction of E. coli-derived kindlin-3). We have previously found kindlin 1-3 to consistently and strongly bind to integrin cytoplasmic tail peptides in pulldown experiments suggesting a lower KD.

One of the main conclusions from the structural data is that kindlin-3 forms homotrimers in vivo. The manuscript unfortunately is a little bit light on the in vivo corroboration of this idea. In fact, I don't think there is any in vivo data presented that would strongly support the idea of kindlin trimers occurring in vivo. A kindlin-3 mutant induces increased adhesion which is in line with the authors hypothesis however no proof that trimers exist in vivo.

Therefore, I was wondering whether the authors are able to generate some in vivo evidence of trimer formation along the lines of FRET or proximity ligation methods? Also, this may help address the question of when, where and how trimers are formed in vivo.

The data presented is definitely worth publishing and it will be up to the editors to decide how much in vivo corroboration is required for this journal. I do think that a little bit more in vivo data would significantly improve the manuscript. They authors talk about developing cancer drugs based on these new findings in the discussion. Prior to taking this leap I would encourage them to further test their idea of homotrimer formation in vivo and make sure that these indeed exist and are important regulators of integrin activity.

Minor points include referring to Kindler syndrome as a skin atrophy disease. This terminology does not exist in dermatology. I would refer to Kindler syndrome as a disorder within the spectrum of hereditary epidermolytic blistering diseases. Likewise I would refer to LAD-III as a primary immunodeficiency with platelet dysfunction.

---------------

Reviewer #3: 

Kindlins play important role in the integrin activation and clustering. Currently the main information on the kindlin structure is based on the fragment with the deleted F2 loop and PH domain. The manuscript for the first time reports the structure of a complete kindlin3. While showing similar overall fold to the truncated version, the structure does not have the F2 domain swap dimerisation. Instead, it forms a trimer where PH domain interacts with the F3 lobe and partially occlude the integrin binding site. In solution kindlin3 exists as a mixture of stable monomer and trimer that can be separated by gel filtration. The trimeric form does not bind integrin, as expected from the crystal structure, while the monomer has a very weak with integrin. Disruption of the trimeric interface by structure-based mutations promotes integrin activation in cells. The authors propose a model where kindlin activity is modulated through trimerisation that is, possibly, disrupted by phosphorylation. Dimerisation reported earlier for the truncated form has not been detected in the full-length kindlin3 and kindlin2 by the authors.

The structure of the full-length kindlin is clearly important for the adhesion field, and the proposed model of kindlin regulation is intriguing. However, the presented experimental support for the regulation model and the role of phosphorylation is rather tentative.

1. The lack of inter-conversion between the monomer and the trimer is puzzling. No structural rearrangement is required, so the equilibrium is expected to be dynamic. The extremely slow inter-conversion suggests that an additional factor may be involved that induces trimer formation. This has an implication for the regulation mechanism that the authors should considered and discuss.

2. The authors concluded that non-phosphorylated form does not bind integrin from the ITC data (Fig S3) and detection of pS484 in sf9, but not in e.coli. The ITC titrations of S3a and d look very similar, except for a larger buffer mismatch in S3a. However, the binding curve of S3d is not fitted correctly, because it shows systematic deviations from the experimental data. The values of the dltH axis of S3d do not seem to correspond to the titration data – the heat changes are similar to those in S3a, so the dltH values are also expected to be similar. The theoretical curve should be fitted correctly and the values checked. From the shape of the curve and the raw data, the binding constant is expected to be similar in S3a and d, which invalidates the conclusion about the effect of phoshorylation on the binding.

3. The ITC experiments should also be conducted on the kindlin2 monomers expressed in e.coli and sf9. The reported integrin interaction is stronger for kindlin2, which will make any potential differences easier to detect.

4. The lack of interaction for e.coli kindlin3 monomer is hard to explain by phosphorylation anyway, because S484 distant from the integrin binding site. Perhaps the real cause of the difference is the C-terminal tag of e.coli construct that somehow reduces the binding because of its proximity to the integrin binding site. According to methods, the tag was not removed. The authors should repeat the binding experiment with the tag removed.

5. MS data do not show how large is the fraction of the phosphorylated protein. This is expected to be too low to have any significant effect on the interactions measured in vitro.

6. The clustering part of the model in fig.5 is a pure speculation and is not based on any of the presented data. It should be removed from the figure as potentially misleading.

7. Description of the preparation of the monomeric and trimeric forms should be included in the methods.

8. The manuscript has a number of somewhat awkward and not fully clear sentences. Further editing and refinement of the text is recommended.

---

## [Decision Letter · Decision Letter 2]

26 Mar 2020

Dear Dr Gao,

Thank you very much for submitting a revised version of your manuscript "Structural basis of human full-length kindlin-3 homotrimer in an auto-inhibited state" for consideration at PLOS Biology. This revised version of your manuscript has been evaluated by the PLOS Biology editors, the Academic Editor and the original reviewers. We apologize again for the delay while our reviewers, Academic Editors and editorial staff deal with COVID-19-related disruptions.

As you can see, Reviewers 2 and 3 are satisfied with the revision and Reviewer 3 has only one further minor request. Reviewer 1 however still has technical concerns and doesn't feel his/her previous points have been adequately addressed. The concerns continue to relate to the physiological relevance of kindlin-3 trimerization and discrepancy with previous work. The Academic Editor has very carefully evaluated these comments. He/she acknowledges that this is a difficult system to study and that previous work (Yates et al., 2012) characterizing full-length mouse Kindlin-3 reports a monomer by SEC and dimer by SAXS. The finding of a homotrimer state via crystal structure is a provocative result and the key title claim of this manuscript. Therefore, the editorial team feel it is essential to have strong and rigorous evidence for this result. We all agree with Reviewer 1 that this is currently not the case. The Academic Editor suggests that definitive support for the trimer state can be obtained by doing SEC-SAXS on a mixture of trimer and monomer and getting an envelope for each. We realize that these are difficult times and you may not be able to perform this experiment quickly due to COVID-19-related shutdowns. We do feel rigorous support for this result is necessary however and are willing to be flexible regarding the revision timeline. I have currently given you 6 months; if you need additional time please let us know. We encourage you to preprint the article (e.g. on bioRxiv) and please also know that your manuscript will fall under our scooping protection policy. Please see below for specific feedback regarding addressing Reviewer 1's comments - 

- First point (discrepancy with Yates et al., 2012) - the Academic Editor notes that the Yates paper is on mouse kindlin-3. And while the expression system is a weak argument, he/she notes that the differential heparin step and/or concentration dependence could be factors. The concerns raised in the first point can be addressed via text discussion.

- AUC has the same caveats as SEC-MALS and SEC, especially since the trimer has a hole in the middle. As noted above, SEC-SAXS would be the best experiment because an envelope of each peak could be made. 

- The Academic Editor agrees with Reviewer 1 that the DSSO experiment is not complete and is unconvincing. Crosslinking should be done with monomer to see if there is some interconversion to trimer, especially since you crystallized the monomer and got a trimer. And a MWM ladder that has a higher MWM than 200 kDa should be used (the current gel is inadequate). 

- The MS data is also weak. It is not clear how many times the crosslink was observed, if this is the only interaction detected, and if so, why. There is also no effort to convince us this is an intermolecular interaction vs intramolecular (by distance arguments), nor whether it is reasonable that 30 Å is OK for a spacer that is 12 Å.

- Regarding the second point (large buried surface area) - this should be addressed directly.

- Other comments-point 1 (regarding the resolution) - we agree with this concern and it should be addressed and updated, as relevant

- Other comments-point 2 - please at least indicate how many times the experiment was done and what type of data is presented (e.g. average).

Overall, in light of the reviews, we will not be able to accept the current version of the manuscript. As noted however, we would welcome re-submission of a revised version that takes into account the reviewers' comments and provides rigorous support for your conclusions. Please note that we cannot make any decision about publication until we have seen the revised manuscript and your response to the reviewers' comments. Your revised manuscript is also likely to be sent for further evaluation by the reviewers.

We expect to receive your revised manuscript within 6 months. Please email us (plosbiology@plos.org) if you have any questions or concerns, or would like to request an extension. At this stage, your manuscript remains formally under active consideration at our journal; please notify us by email if you do not intend to submit a revision so that we may end consideration of the manuscript at PLOS Biology.

**IMPORTANT - SUBMITTING YOUR REVISION**

1. A 'Response to Reviewers' file - this should detail your responses to the editorial requests noted above, present a point-by-point response to all of the reviewers' comments, and indicate the changes made to the manuscript. 

*Re-submission Checklist*

*Published Peer Review*

*PLOS Data Policy*

*Blot and Gel Data Policy*

Sincerely,

Hashi Wijayatilake, PhD, 

Managing Editor

PLOS Biology

REVIEWS:

Reviewer #1: 

The authors made effort to address some of my concerns in the revised manuscript. Unfortunately, the major concerns were not addressed and I remain unconvinced about the physiological relevance of kindlin-3 trimerization, which is the key conclusion/major selling point of the study. In particular, the authors failed to define the minor peak as a trimer in their gel filtration experiment, and such minor peak was not observed in other previous studies. Even if they approved this minor peak were a trimer, the low population of such “trimer” vs monomer would still question the physiological relevance of the trimer. My primary concerns are as follows:

First, as I mentioned last time, Yates et al., 2012 only observed kindlin-3 monomer using the similar insect cell expression system. Structurally Yates et al also only observed the monomer. In previous version, the authors ignored this apparent contradiction and simply explained the trimerization they observed in insect cells but not in E. coli was due to the expression condition. In this revised version, the authors argued that the purification protocols are different in the two studies. This is very weak argument. Yates et al purification protocol is standard! If their conditions could not reveal a trimer, I would strongly suspect whether the authors’ experimental conditions generated some non-physiological oligomer in the minor peak position, which the authors claim as a trimer in order to support their crystal structure. Please note that even based on the authors’ purification condition, the monomer is predominant from the gel filtration. The so-called “trimer” might be an oligomer or aggregation depending on the injected protein concentration and some unknown reasons such as disulfide-bond induced cross linking. I say this because the size exclusion chromatography largely depends on stokes radius of molecules that gives uncertain molecular weight (MW) information. Thus, the SEC-MALS (Figure 4c) may not always provide conclusive evidence but limited information for estimating apparent MW of proteins. One cannot eliminate the possibility of the observed minor peak as a dimer or tetramer or an oligomer! Indeed, in the newly generated figure of size exclusion chromatography (Fig. 4a,b), the elution positions of MW standards do not seem to match. More quantitative assessments such as analytical ultracentrifugation (AUC) are needed for characterizing this minor peak. Even if the authors can approve this is a trimer in vitro, the physiological relevance of the minor population of such trimer is questionable.

Second, the human kindlin-3 utilized for crystallization was a monomer as the authors described in the main text. This strongly indicates that the trimer they observed in the crystal lattice is due to the crystal packing. This kind of phenomenon was already previously observed in mouse kindlin-2 that began with monomeric form during crystallization but was crystallized as a dimer (Li et al., PNAS, 2017). Consistent with my view of the crystallization artifact, the large buried surface area (~1,850 A^2) in the trimer interface is huge. Such huge interface would lead to a very tight trimer as a major form. By contrast, the authors observed the monomer predominantly and so-called “trimer” was a very minor peak in the size exclusion experiment. This again indicates that the so-called “trimer” in the crystal structure is physiologically irrelevant, which is clearly a crystallization artifact.

Other comments:

1), the crystallographic data collection statistics is still poorly characterized. The authors failed to incorporate the Rmerge/R-meas in the data collection statistics table that is essential component for the main table in crystal structure report. Demonstration of all the essential components in the data collection and refinement statistics in the main table is crucial. The R-meas of over 500% in outer shell (500% discrepancy for the measured intensities!) are not acceptable by standard criteria. The resolution limit would be thus around at 4.3-A according to their statistics. The authors need to more carefully reevaluate the resolution limit followed by structure refinement. The authors also need to state the number of native data sets to be merged in the method section. 

2), the ITC measurement for the binding of kindlin-2/-3 to integrin has been criticized by all three reviewers but the authors did not seem to reply to their concerns for distinct binding affinity between two isoforms of kindlin and technical viewpoint. No saturation point was observed for the binding between monomeric kindlin-3 and integrin (Fig. S9a), but was observed for monomeric kindlin-2 (~200 uM vs. 13 uM). 

3). Fig. S10 seems to be the overlay of two structures instead of three as described in legendary.

4). Line 540, typo, grid. There are several other typographical errors and uncertain sentences in the revised text. 

--

Reviewer #2: 

The authors made a nice effort to address my previous concerns. I am fully satisfied with their approach at this point and recommend the paper for publication.

--

Reviewer #3: 

The authors made large changes to the manuscript in response to the comments. The rather speculative at this stage phosphorylation part of the study is completely removed, making the paper clear and focused. Now the story is straightforward: trimers are observed by crystallography and detected in cells; trimers do no bind integrin; mutants disrupting trimer formation enhance intergrin activation; therefore, trimer formation may be a mechanism of kindlin activity regulation and should be considered in the functional models. In addition, dimers that were previously proposed as important, have not been detected and may have been an artefact of truncation in the previous studies. In my view these are important messages for the field which may lead to a revised functional model of kindlin activity.

All my comments have been addressed, mainly by the removal of more speculative or poor quality data. Writing now is clear an easy to follow.

The only surprising omission is the lack of ITC data for K2 trimer. This should be fairly simple to measure as the authors have both monomers and trimers of K2. I suggest that these data are included in the manuscript.

---

## [Editor Report · Decision Letter 3]

11 May 2020

Dear Dr Gao,

Thank you for submitting your revised Research Article entitled "Structural basis of human full-length kindlin-3 homotrimer in an auto-inhibited state" for publication in PLOS Biology. We have now discussed the revision with the Academic Editor. We appreciate your addition of more analytical details, including monomer cross linking that shows a trimer and a SEC trace of the results with and without monomer cross linking. We are willing to accept some of the rebuttals and responses provided to the other requests. However, in order to proceed, we will need the raw mass spec data provided, and the related analyses should be presented directly in the paper (e.g. MS spectrum of crosslinked peptide, corresponding fragmentation spectrum, fragment mass match accuracy, ion assignments). 

Please also address the following points:

> Data Availability Statement: Please update your Data Availability Statement with crystal structure deposition information and code/s. The Materials and Methods section has the following statement: "Coordinates and structure factors have been deposited in the Protein Data Bank with accession code 6AEN" - please also duplicate this in the Data Availability Statement.

> This PDB 6AEN dataset also needs to be publicly released (currently 'on hold')

> As noted above, please provide the MS analysis data. Here are the recommended repositories for this type of data:

https://journals.plos.org/plosbiology/s/recommended-repositories

https://massive.ucsd.edu/ProteoSAFe/static/massive.jsp

> Please also make sure to address the other numerical data and other policy-related requests noted at the end of this email.

We expect to receive your revised manuscript within two weeks. Your revisions should address the specific points made by each reviewer. In addition to the remaining revisions and before we will be able to formally accept your manuscript and consider it "in press", we also need to ensure that your article conforms to our guidelines. A member of our team will be in touch shortly with a set of requests. As we can't proceed until these requirements are met, your swift response will help prevent delays to publication.

*Copyediting*

*Published Peer Review History*

*Early Version*

*Submitting Your Revision*

Sincerely,

Hashi Wijayatilake, PhD, 

Managing Editor

PLOS Biology

DATA POLICY:

Figs. 2D, 4ABC, 5C, 6BC, S7B, S9, S10

**As noted above, please also provide the MS analysis data. Here are the recommended repositories for this type of data:

https://journals.plos.org/plosbiology/s/recommended-repositories

https://massive.ucsd.edu/ProteoSAFe/static/massive.jsp

**Please also ensure that figure legends in your manuscript include information on where the underlying data can be found, and ensure your supplemental data file/s has a legend.

**Please ensure that your Data Statement in the submission system accurately describes where your data can be found.

PLOS Biology requires the original, uncropped and minimally adjusted images supporting all blot and gel results reported in an article's figures or Supporting Information files. We will require these files before a manuscript can be accepted so please prepare and upload them now. Please carefully read our guidelines for how to prepare and upload this data: https://journals.plos.org/plosbiology/s/figures#loc-blot-and-gel-reporting-requirements

---

## [Editor Report · Decision Letter 4]

22 Jun 2020

Dear Dr Gao,

On behalf of my colleagues and the Academic Editor, Raquel L. Lieberman, I am pleased to inform you that we will be delighted to publish your Research Article in PLOS Biology. 

Early Version

PRESS 

Kind regards,

Alice Musson

Publishing Editor, 

PLOS Biology

on behalf of

Di Jiang, PhD,

Senior Editor

PLOS Biology